# Improvements on Uncertainty Quantification for Node Classification via Distance-Based Regularization

**Russell Alan Hart**
The University of Texas at Dallas
rah150030@utdallas.edu

**Linlin Yu**
The University of Texas at Dallas
linlin.yu@utdallas.edu

**Yifei Lou**
University of North Carolina at Chapel Hill
yflou@unc.edu

**Feng Chen** *
The University of Texas at Dallas
feng.chen@utdallas.edu

## Abstract

Deep neural networks have achieved significant success in the last decades, but they are not well-calibrated and often produce unreliable predictions. A large number of literature relies on uncertainty quantification to evaluate the reliability of a learning model, which is particularly important for applications of out-of-distribution (OOD) detection and misclassification detection. We are interested in uncertainty quantification for interdependent node-level classification. We start our analysis based on graph posterior networks (GPNs) that optimize the uncertainty cross-entropy (UCE)-based loss function. We describe the theoretical limitations of the widely-used UCE loss. To alleviate the identified drawbacks, we propose a distance-based regularization that encourages clustered OOD nodes to remain clustered in the latent space. We conduct extensive comparison experiments on eight standard datasets and demonstrate that the proposed regularization outperforms the state-of-the-art in both OOD detection and misclassification detection.

## 1   Introduction

In recent years, deep neural networks (DNNs) have been widely used in various fields [10, 28]. However, some neural networks provide under-confident [36] or over-confident [14] predictions, limiting their practical applications in risk-constrained and safety-critical fields, such as drug discovery [38], autonomous driving [30], and medical diagnosis [2]. Take autonomous drug design for an example. Uncertainty estimation on the reliability of model predictions helps to support molecular reasoning and experimental design by saving considerable time and resources [23]. It is important to estimate the predictive uncertainty of a DNN, i.e., indicating when its predictions are likely incorrect. There are two main types of uncertainty: epistemic uncertainty (knowledge uncertainty) and aleatoric uncertainty (data uncertainty) [9]. Epistemic uncertainty is due to the lack of knowledge about unseen data. Aleatoric uncertainty is caused by the inherent complexity of the data, which cannot be reduced by increasing the training data, including sources of noise such as homoscedastic or heteroscedastic noise [17]. These two uncertainty types are typically used for out-of-distribution (OOD) and misclassification detection, respectively.

Most models have been introduced for uncertainty estimation on i.i.d. inputs, such as image and tabular data. However, the uncertainty estimation for classifying interdependent nodes in attributed graph data, such as social networks and citation networks, is under-explored. This work focuses on the node classification tasks with great potential to generalize to others with interdependent inputs.

---

*corresponding author

37th Conference on Neural Information Processing Systems (NeurIPS 2023).

Among various graph neural networks (GNNs) for processing graph data structures [18, 35, 13, 8], graph posterior network (GPN) has been developed for semi-supervised node classification tasks [32] that achieves state-of-the-art results in uncertainty estimation.

The major **contributions** of this paper are three-fold: (1) We theoretically analyze the limitations of GPN at OOD detection when minimizing uncertainty cross-entropy (UCE), a widely used loss function for uncertainty estimation. (2) Motivated by the aforementioned limitations, we propose a distance-based regularization that considers the prior knowledge that OOD-specific features are useful for learning representational space mappings. (3) We conduct extensive experiments comparing our proposed model with five state-of-the-art baselines on eight graph datasets for two uncertainty quantification tasks: OOD detection and misclassification detection tasks. The results demonstrate that our proposed regularization can improve the quality of uncertainty quantification.

## 2   Related Work

This section reviews existing uncertainty estimation methods for i.i.d data and graph data.

**Uncertainty Quantification for i.i.d inputs** – There is plentiful research on uncertainty quantification on i.i.d. inputs as discussed in a recent survey [1]. The first family quantifies the predictive uncertainty of a DNN via multiple forward passes, such as deep ensembles and dropout-based Bayesian neural networks (BNNs). Deep ensembles [19] intuitively sample multiple predictions by training an ensemble of deep neural networks and aggregate the results. Dropout-based methods [11] utilize multiple stochastic forward passes implemented with different dropout initializations to approximate the posterior distribution of network weights. However, the substantial memory and computational demands required for training and testing make it impractical for real-time applications. The second family quantifies uncertainty using deterministic single forward-pass neural networks, including density-based methods and distribution-based methods. The density-based approaches typically fit a distribution (e.g., class-wise Gaussian distribution [3, 20, 34]) in the representation space of a pre-trained or fine-tuned DNN, followed by the associated PDF function to quantify different uncertainty types. The distribution-based methods train a deterministic neural network that directly predicts the conjugate prior distribution of the class probabilities of the input feature vector, called Dirichlet distribution, for uncertainty quantification. The predicted Dirichlet distribution can be interpreted as an approximation of the posterior distribution of class probabilities conditioned on the input feature vector. Popular distribution-based models are prior networks [22], evidential networks [29], and posterior networks (PN) [7], all using UCE as the loss function with different regularizations to improve the quality of uncertainty quantification.

**Uncertainty Quantification for Graphs** – As pointed out in the survey [1], uncertainty quantification on GNNs and semi-supervised learning is under-explored. Most existing models for uncertainty quantification on graphs are either dropout-based or BNN-based methods that typically drop or assign probabilities to edges. There are two approaches using deterministic single-pass GNNs to quantify uncertainty. One is called graph-based kernel Dirichlet distribution estimation (GKDE) [39], which consists of evidential GCN, graph-based kernel, teacher network, dropout, and loss regularization. Another method is the GPN model that combines PN and personalized page rank (PPR) message passing to disentangle uncertainty with and without network effects. In addition, a recent method [37] used standard classification loss for OOD detection on graphs together with an energy function that is directly extracted from GNN, however, it is limited to OOD detection, not generally on the topic of uncertainty quantification.

## 3   Preliminary

We discuss the problem setting of uncertainty quantification on the task of semi-supervised node classification in Section 3.1. In particular, we use a deep neural network to predict the multinomial uncertainty for each node and evaluate the aleatoric uncertainty and epistemic uncertainty by the prediction result. In Section 3.2, we give a brief review of the GPN model [32], which serves as a fundamental framework for our analysis and motivation for the proposed approach.

## 3.1 Problem Setting

We define a graph with attributed node-level features $\mathcal{G} = (\mathcal{V}, \mathcal{E}, X, Y_{\mathbb{L}})$, where $\mathcal{V}$ is a set of nodes on the graph with cardinality $N$ and $\mathcal{E} \subset \mathcal{V} \times \mathcal{V}$ denotes a set of graph edges that can be represented by an adjacency matrix $W$. A feature matrix is denoted by $X = [\mathbf{x}_1, \ldots, \mathbf{x}_N]^T \in \mathbb{R}^{N \times d}$, in which each row $\mathbf{x}_i \in \mathbb{R}^d$ is a feature vector of node $i$ with dimension $d$. Under the semi-supervised learning setting, a set of labels is available, denoted by $Y_{\mathbb{L}} = \{y_i \mid i \in \mathbb{L}\}$, where $\mathbb{L} \subset \mathcal{V}$ and $y_i \in \{1, \ldots, K\}$ for $K$ classes.

Our goal is to design and learn a deterministic GNN based on $\mathcal{G}$ that takes the feature matrix $X$ and the adjacency matrix $W$ as INPUT and predicts the parameters of a Dirichlet distribution for each node $i \in \mathcal{V}$ as OUTPUT, denoted as $\boldsymbol{\alpha}_i$, which is often referred to as the concentration parameters. Therefore, the network function $F_{\Theta}$ can be expressed as: $\mathcal{A} = F_{\Theta}(X, W)$, where $\mathcal{A} := [\boldsymbol{\alpha}_i]_{i \in \mathcal{V}}$ is a matrix and $\Theta$ refers to network parameters. The statistical relations between the class label $\mathbf{y}_i$, the vector of class probabilities $\mathbf{p}_i$, and the Dirichlet parameters $\boldsymbol{\alpha}_i$ can be represented as:

$$\mathbf{y}_i | \mathbf{p}_i \sim \text{Cat}(\mathbf{p}_i), \quad \mathbf{p}_i | \boldsymbol{\alpha}_i \sim \text{Dir}(\boldsymbol{\alpha}_i), \quad [\boldsymbol{\alpha}_i]_{i \in \mathcal{V}} = F_{\Theta}(X, W). \tag{1}$$

Based on the predictions of $\mathcal{A}$, the expected vector of class probabilities $\bar{\mathbf{p}}_i := \mathbb{E}[\mathbf{p}_i | \boldsymbol{\alpha}_i] = [\alpha_{i1}/\alpha_{i0}, \cdots, \alpha_{iK}/\alpha_{i0}]^T$, where $\alpha_{i0} = \sum_{k=1}^{K} \alpha_{ik}$ is called the Dirichlet strength. The aleatoric and epistemic uncertainties about the classification of each node $i$ can be calculated as:

$$u_i^{\text{alea}} = -\max\{\bar{p}_{i1}, \cdots, \bar{p}_{iK}\} \quad \text{and} \quad u_i^{\text{epis}} = -\alpha_{i0}, \tag{2}$$

respectively. The aleatoric uncertainty is measured by the negative of the largest class probability in $\bar{\mathbf{p}}_i$. This uncertainty is higher when the largest class probability in $\bar{\mathbf{p}}_i$ is lower, which implies that the model is less confident and the probabilities are more evenly spread across classes. On the other hand, the epistemic uncertainty is measured by the negative of the Dirichlet strength $\alpha_{i0}$, whose value is higher when the Dirichlet strength is lower, meaning that the model is unfamiliar with the feature vector of node $i$ and the predicted Dirichlet distribution is less concentrated around a specific point or set of points on the probability simplex [22]. We note that a high aleatoric uncertainty may not indicate a high epistemic uncertainty and vice versa. For example, two evidence parameters $\boldsymbol{\alpha}_i = [1, \cdots, 1]$ and $\boldsymbol{\alpha}_j = [1000, \cdots, 1000]$ have the same aleatoric uncertainty: $-1/K$, since they have the same projected class probabilities; but their epistemic uncertainties differ drastically: $K$ versus $1000K$. Please refer to [32, 33] for rationales of aleatoric and epistemic uncertainties in (2).

## 3.2 Graph Posterior Network

Our framework is based on graph posterior network (GPN) [32], which extends posterior network (PN) [7] to semi-supervised node classification. GPN consists of three main steps. First, a feature encoder maps the original features onto a low-dimensional latent space with a simple two-layer multi-layer perception (MLP) encoder. Second, a radial normalizing flow [27] estimates the density of the latent space per class. Lastly, a personalized page rank message passing scheme [13] diffuses the pseudo counts (density multiplied by the number of training nodes) by taking the graph structure into account. We summarize the three steps with notations as follows,

1. Multi-layer perceptron for representation learning: $\mathbf{z}_i = f(\mathbf{x}_i; \boldsymbol{\theta})$ or $f_{\boldsymbol{\theta}}$ in short.

2. Normalizing flow for density estimation: $g_{\boldsymbol{\phi}}$ for short, and more specifically

$$g_{\boldsymbol{\phi}}(\mathbf{z}_i)_k = N_k \cdot \mathbb{P}(\mathbf{z}_i | k; \boldsymbol{\phi}), \tag{3}$$

where $N_k$ is the number of training nodes belonging to the class $k$, $\mathbf{z}_i$ is the embedding vector of node $i$ obtained via the first step, $\mathbb{P}(\mathbf{z}_i | k; \boldsymbol{\phi})$ is the conditional density per class $k$ estimated by a normalizing flow module, and $\boldsymbol{\phi}$ denotes the parameters of this module. GPN also includes the evidence computed prior to the graph aggregation, defined by

$$\boldsymbol{\alpha}_i^{\text{feat}} = g_{\boldsymbol{\phi}}(\mathbf{z}_i) + \mathbf{1}. \tag{4}$$

3. Personalized page rank (PPR) for evidence diffusion: $\beta_{i,k}^{\text{aggr}} = \sum_{j \in \mathcal{V}} \mathbf{\Pi}_{i,j}^{\text{ppr}} g_{\boldsymbol{\phi}}(\mathbf{z}_j)_k$, where $\mathbf{\Pi}_{i,j}^{\text{ppr}}$ refer to the dense PPR scores implicitly reflecting the importance of node $j$ from the perspective of node $i$. Then we can get the predicted concentrate parameters $\boldsymbol{\alpha}$ with a uniform prior $\mathbf{1}$ for a non-degenerated Dirichlet distribution, i.e.,

$$\boldsymbol{\alpha}_i = \beta_i^{\text{aggr}} + \mathbf{1}. \tag{5}$$

As opposed to GPN, PN is designed for uncertainty estimation for i.i.d. inputs, which only considers the first two steps to predict the Dirichlet distribution $\mathrm{Dir}(\boldsymbol{\alpha}_i^{\mathrm{feat}})$.

Given the labels of training nodes: $Y_{\mathbb{L}} = \{y_i \mid i \in \mathbb{L}\}$, GPN is trained by minimizing the following Bayesian loss:

$$\mathcal{L} = \mathrm{UCE}(\mathcal{A}, Y) + \lambda \sum_{i \in \mathbb{L}} \mathbb{H}[\mathrm{Dir}(\boldsymbol{\alpha}_i)]. \tag{6}$$

The first term in (6), called uncertainty cross entropy (UCE) [5], is defined by

$$\mathrm{UCE}(\mathcal{A}, Y) = \sum_{i \in \mathbb{L}} \mathbb{E}_{\boldsymbol{p}_i \sim \mathrm{Dir}(\boldsymbol{\alpha}_i)} \left[ -\log \mathbb{P}(y_i | \mathbf{p}_i) \right] = \sum_{i \in \mathbb{L}} \sum_{k \in [K]} y_{ik} (\Psi(\alpha_{i0}) - \Psi(\alpha_{ik})), \tag{7}$$

where $Y = [\mathbf{y}_i]_{i \in [N]}$, $\mathbf{y}_i \in \{0, 1\}^K$ is the one-hot encoded ground-truth class of the node $i$, and $\Psi$ is the digamma function, in terms of the Gamma function by: $\Psi(x) = \frac{\Gamma'(x)}{\Gamma(x)}$. Minimizing UCE is known to increase confidence in classifying observed data (training nodes in this context). The second term in (6) is based on the entropy of each node-level Dirichlet distribution $\mathrm{Dir}(\boldsymbol{\alpha}_i)$ that favors smooth distributions. For more details on GPN, please refer to Appendix C.

# 4 Our Contributions

This section provides a series of theoretical analyses relating to the UCE loss term and the GPN model for detecting the OOD nodes, followed by a partial remedy to derived issues via two distance-based regularizations. Specifically, we prove in Theorem 1 that under certain conditions, UCE can be made arbitrarily small with the limiting case of UCE equal to zero in Corollary 3. Theorem 4 gives a construction to make the UCE to be zero. As UCE does not involve the OOD nodes, Theorem 6 and Corollary 7 elucidate scenarios for possibly detecting the OOD nodes. Lastly, Theorem 8 presents a special situation where GPN fails to detect the OOD nodes.

## 4.1 Theoretical analysis

The loss function plays a pivotal role in learning effective representation functions and density estimations. In this context, we establish several theorems (Theorems 1 and 4) to describe some demanding assumptions on $f_\theta$ and $g_\phi$ that achieve the minimum UCE loss. We then describe a limitation of UCE in separating ID and OOD nodes in Theorem 6 and Corollary 7 for PN, which means that we only consider the first two steps in GPN. The main conclusion of our analysis is that the UCE loss function alone is insufficient to learn a representation space that separates OOD from ID nodes. We take graph connectivity into account in Theorem 8 to study some scenarios where GPN is ineffective for OOD detection. Although our theorems do not completely characterize graph learning, they provide some insights into the behavior of the network parameters in PN/GPN when minimizing the UCE loss.

**Theorem 1.** *If the underlying distribution of feature vectors belonging to class $k$, denoted by $\mathcal{X}_k$, is disjoint to each other and both the MLP module ($f_{\boldsymbol{\theta}}$) and the normalizing flow module ($h_{\boldsymbol{\phi}}$) can be arbitrarily complex, then $\forall \epsilon > 0$ there exists a configuration of $f_\theta$ and $g_\phi$ such that $UCE(\mathcal{A}, Y) < \epsilon$.*

For the proofs, please refer to Appendix B. Here, we elaborate on the ideal configuration that satisfies the conclusion of Theorem 1. We assume the MLP function $f_\theta$ is arbitrarily complex such that it maps $\mathbf{x}_i \in \mathcal{X}_k$ into a bounded ball in the representational space, i.e.,

$$\{f_\theta(\mathbf{x}_i) | i \in [N] \text{ and } \mathbf{x}_i \in \mathcal{X}_k\} \subset B(\mathbf{z}_k, r_k),$$

where each ball $B(\mathbf{z}_k, r_k)$ is centered at a point $\mathbf{z}_k$ and bounded in size with $r_k < r$ for a positive value $r$. Furthermore, we choose the normalizing flow $g_\phi$ to be

$$g_\phi(\mathbf{z}; k) = \begin{cases} \frac{1}{\mathrm{Vol}(B(\mathbf{z}_k, r_k))} & \text{if } d(\mathbf{z}, \mathbf{z}_k) < r_k \\ 0 & \text{otherwise}, \end{cases} \tag{8}$$

where $\mathrm{Vol}(\cdot)$ refers to the volume of the ball. The conclusion in Theorem 1 states that for every $\epsilon$, there exists a suitable upper bound $r$ of all the balls such that $UCE(\mathcal{A}, Y) < \epsilon$.

An implication of Theorem 1 is that UCE is not sufficient to separate OOD from ID nodes. Example 2 illustrates a scenario that OOD nodes can be close to ID nodes in the learned representation space even though they can be separated in the feature space based on OOD-specific features, which are unfortunately discarded. In other words, the learned representation space by GPN based on the UCE loss is not guaranteed to preserve the distance between OOD and ID nodes in its representation learning step.

**Example 2** (Lost Features). *Suppose two ID classes in a citation network contain bags of words for SVM and neural networks papers respectively. Additionally, the OOD nodes contain bags of words from reinforcement learning papers. Note that frequencies of keywords are used to discriminate different classes. Then the keywords, "actor critic" and "policy network" are able to separate OOD nodes from ID nodes, but are irrelevant features for discriminating between the two ID classes. UCE, as a discriminatory loss, is only applied on ID nodes, and hence it is almost impossible to learn representations that respect the OOD-specific features such as "actor-critic" and "policy networks".*

**Limitations and discussions** $-$ The first assumption in Theorem 1 regarding the distinct class-specific distributions of feature vectors might not be realistic in practice since certain ID classes may not be clearly distinguishable due to noise in features or class labels. Nevertheless, our intuition suggests that if the UCE loss is inadequate for separating OOD from ID nodes in situations where they are separable, it is even more likely to falter in the more complex, non-separable cases. As for the second assumption in Theorem 1, it is true that arbitrary complexity of $f_\theta$ and $g_\phi$ does not fully respect the inductive bias [24] of the network design, such as the MLP layers with ReLU for the feature encoder $f(\mathbf{x}; \boldsymbol{\theta})$, our analysis remains insightful and informative about the structures these networks are likely to exhibit. For example, we may expect from Theorem 1 that the representation of each class may favor an embedded space that compresses OOD-specific features, while density estimation $g_\phi$ tends to have higher peaks over smaller volumes as the model consolidates the representation space.

We note that in [7, Theorem 1] and [32, Theorem], the authors demonstrated that PN/GPN is able to achieve reasonable uncertainty estimation when the feature encoder is a ReLU network and PPR diffusion is removed to disregard network effects. Unfortunately, the analysis is based on extreme node features, specifically as $\delta \to \infty$,

$$\mathbb{P}(f(\delta \cdot \mathbf{x}_i; \boldsymbol{\theta})|k; \boldsymbol{\phi}) \to 0, \quad \text{and} \quad \beta_{i,k}^{\text{agg}} \to 0,$$

for any node $i$ with a high probability. Moreover, Theorem 1 in the GPN paper [32] holds even when the supports of disjoint ID and OOD classes in the latent space overlap. In other words, this result does not prevent distant nodes from having similar representations. In summary, the analysis based on extreme node features may provide limited insights about the issues of GPN studied in this section. See Appendix D for detailed discussions.

By taking $\epsilon \to 0$, Theorem 1 reduces to Corollary 3 where UCE is equal to 0.

**Corollary 3.** *In the ideal case, where the representation function $f_\theta$ maps the support of each class, $\mathcal{X}_k$, to a countable set (with measure zero), $\mathcal{Z}_k$ and there exists a normalizing flow that has infinite density on the point set for every class, one achieves $UCE(\mathcal{A}, Y) = 0$.*

Next, we aim for the construction of a specific case to make UCE equal to zero. As it is challenging to analyze the joint minimization on $\theta$ and $\phi$, we assume that the normalizing flow can be chosen optimally. For this purpose, we consider a simplified problem where the true density function is assumed to be known for a given $\theta$ and hence it can be used to replace the learning of the normalizing flow. As a result, the problem reduces to the learning of the representation network $\theta$.

**Theorem 4.** *Let $\mathcal{X}_k$ be the true distributions for class-$k$ in the original feature space. Suppose the normalizing flow module $g_\phi$ obtains the true analytic solution. If the true distributions $\{\mathcal{X}_k\}$ are disjoint, then the $f_{\hat{\theta}}$ that minimizes the UCE loss projects the support of each class in the original space to a disjoint point set $\mathcal{Z}_k$, where $\mathcal{Z}_k$ is defined by the projection of $\mathcal{X}_k$ to the representation space, i.e., $\mathcal{Z}_k = f_\theta(\mathcal{X}_k)$.*

Notice that the true analytical solution $\hat{\phi}$ in Theorem 4 is a function of $\theta$, i.e., $\hat{\phi} = \phi(\theta)$. It is possible to know an analytical form of $\phi(\theta)$. For example, if the data points belonging to each class are sampled from a known Gaussian distribution in the original feature space and the representation network is a linear projection function, then the true density of the projected data points belonging to each class can be derived based on any configuration of the known Gaussian distribution in the original feature space.

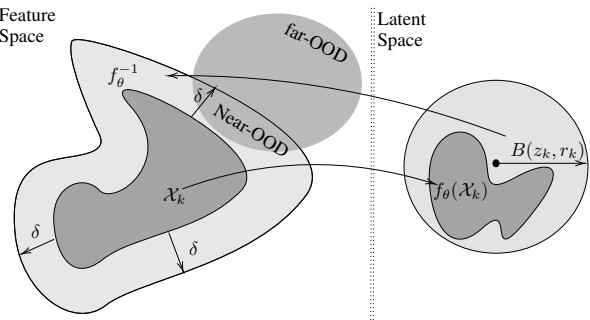

Figure 1: Illustration of the representation mapping in Theorem 1 with the conditions to detect far OOD nodes (Theorem 6) and near OOD nodes (Corollary 7).

Before discussing OOD detection, we start with the definition of OOD nodes.

**Definition 5.** *We define out-of-distribution (OOD) nodes to be the nodes that do not belong to any of the $K$ in-distribution (ID) classes. We denote all the ID distribution supports by $\mathcal{X}_{[K]} = \bigcup_{k=1}^{K} \mathcal{X}_k$.*

In order to detect the OOD nodes, we need a good representation, e.g., $f_\theta(\mathcal{X}_{[K]}) \bigcap f_\theta(\mathcal{X} - \mathcal{X}_{[K]}) = \varnothing$. However, it is possible that $\cup_{k=1}^{K} f_\theta^{-1}(\mathcal{Z}_k) = \mathcal{X}$, which implies that any OOD node in the feature space is mapped to the same distribution of an ID class in the representation space. To this end, we require the preimage, $f_\theta^{-1}$, to be well-behaved in the sense that $f_\theta^{-1}(\mathcal{Z}_k)$ must be contained within a bounded region near the support of the true distribution $\mathcal{X}_k$. Explicitly, we require there exists a constant $\delta > 0$ such that $d(\mathbf{x}, \hat{\mathbf{x}}) < \delta, \forall \hat{\mathbf{x}} \in f_\theta^{-1}(\mathcal{Z}_k)$ and $\forall \mathbf{x} \in \mathcal{X}_k$ for each $k$. The choice of $\delta$ depends on additional information about the dataset. An excessively large $\delta$ causes overlap between classes, thus increasing the likelihood of improperly identifying the OOD nodes as ID. On the other hand, a relatively small $\delta$ forces the model to overfit, which labels the non-training nodes as OOD.

We characterize in Theorem 6 that under certain conditions, OOD can be detected correctly if they are far away from the ID distribution. One such condition is that the function $f_\theta$ is well-fit, meaning that it maps the support of $\mathcal{X}_k$ inside $B(\mathbf{z}_k, r_k)$ for every $k$ in $[K]$. We denote a set of all well-fit functions by $\mathbf{\Gamma}$. Please refer to Figure 1 for a geometric illustration of these relevant quantities.

**Theorem 6** (Far OOD). *Under the same assumptions in Theorem 4 and two additional assumptions: (i) $\mathcal{X}_k$ is bounded and (ii) for any $\hat{\mathbf{x}} \in f_\theta^{-1}(\mathbf{z})$ with $\mathbf{z} \in \mathcal{Z}_k$, there exists $\mathbf{x}_k \in \mathcal{X}_k$ such that $d(\mathbf{x}_k, \hat{\mathbf{x}}) < \delta$, if the true distribution of OOD nodes does not overlap with the region $\cup_{\theta \in \mathbf{\Gamma}} \cup_k \cup_{z_k \in \mathcal{Z}_k} \{f_\theta^{-1}(z_k)\}$, then minimizing UCE can learn a representation projection of $f_\theta$ that detect all the OOD points farther than $\delta$ from $\mathcal{X}_k$.*

**Corollary 7** (Near OOD). *Following Theorem 6, if $\hat{\mathbf{x}} \in \cup_{\theta \in \mathbf{\Gamma}} \cup_k \cup_{z_k \in \mathcal{Z}_k} \{f_\theta^{-1}(z_k)\}$ follows the distribution of OOD nodes, then the classification of $\hat{\mathbf{x}}$ depends on the choice of $\theta \in \mathbf{\Gamma}$.*

The main purpose of Theorem 6 is to show a desired behavior for OOD detection is induced by point-wise effects. In practice, we suggest the careful construction of $f_\theta$'s topology coupled with proper regularization terms to achieve the point-wise effect.

Lastly, we consider the setting of GPN to use a graph layer after the feature-wise evidence predictions.

**Theorem 8.** *Under the following conditions: (1) The features of some class 0 and OOD nodes belong to $S$; (2) The OOD nodes are only connected to class 0 and themselves; (3) Other nodes belong to other regions i.e. $\mathbf{x} \in X - S$ if $\mathbf{x} \notin \mathcal{X}_0 \bigcup \mathcal{X}_{OOD}$; (4) Other nodes' features are non-degenerate in their associated region; and (5) the endowed graph neural network layer must produce evidence for each node between the highest and lowest feature evidence found among its neighbors. A graph with arbitrary homophily can achieve a global minimum on UCE with the associated OOD nodes achieving arbitrarily large evidence, while simultaneously having perfect accuracy.*

**Limitations and Discussions.** In Theorem 8, we provide a special situation where the GPN architecture may fail to detect OOD nodes by predicting large evidence values for belonging to class 0. In addition, we show in Appendix B that if GPN has bad initial feature predictions, even ideal graph construction coupled with an ideal graph neural network (with a homophily degree 1.0) fails to

prevent the OOD nodes from being misclassified as ID nodes. For homophily graphs with degrees less than 1.0, the majority of nodes for each class have similar evidence values after graph diffusion layers, except that the nodes between some pairs of classes may have different evidence values.

## 4.2 Distance-Based Regularization

As discussed in Section 4.1, the UCE loss function alone is insufficient to learn a representation space that separates OOD from ID nodes using the GPN model. We propose a heuristic remedy that enforces distance minimization on the graph. Ideally, we should design a distance formula that can preserve the distance relationship among all the feature vectors. However, distance preservation likely increases variation in the latent space as we cannot compress the classes' support in the representation space to be arbitrarily small, while simultaneously preserving distances.

Instead, we consider distance minimization, as it helps prevent the model from discarding relevant features while decreasing variation between nodes in the representation space. Here we give two formulations directly. The simpler, theorem-motivated term, is the distance regularization on the latent space,

$$\mathrm{R}_D(f_\theta(X); \mathcal{G}) = \sum_{i,j \in \mathcal{E}} \|f_\theta(\mathbf{x}_i) - f_\theta(\mathbf{x}_j)\|^2. \tag{9}$$

The regularization encourages nearby points in the graph representation to remain nearby in the latent space. In other words, this regularization (9) discourages the overlap $f_\theta(\mathcal{X}_{[K]}) \bigcap f_\theta(\mathcal{X} - \mathcal{X}_{[K]})$.

We also minimize the "distance" on the produced evidence through a divergence-based regularization,

$$\mathrm{R}_\alpha(\boldsymbol{\alpha}^{\mathrm{feat}}; \mathcal{G}) = \sum_{(i,j) \in \mathcal{E}} \mathrm{div}_{\mathrm{kl}}(\boldsymbol{\alpha}_i^{\mathrm{feat}}, \boldsymbol{\alpha}_j^{\mathrm{feat}}) + \mathrm{div}_{\mathrm{kl}}(\boldsymbol{\alpha}_j^{\mathrm{feat}}, \boldsymbol{\alpha}_i^{\mathrm{feat}}), \tag{10}$$

where $\mathrm{div}_{\mathrm{kl}}$ refers to the Kullback–Leibler divergence and two symmetric terms are considered. This divergence-based formulation likely decreases the variation in evidence between neighboring nodes, because high variation in the latent space between neighboring nodes need not be mapped to similar evidence.

In summary, we augment the GPN mode with either one of the proposed regularization terms in (9) and (10), thus leading to the objective function as follows,

$$\mathcal{L} = \mathrm{UCE}(\mathcal{A}, Y) - \lambda_1 \underbrace{\sum_{i \in \mathbb{L}} \mathbb{H}(\mathrm{Dir}(\boldsymbol{\alpha}_i))}_{\text{entropy regularizer}} + \lambda_2 \underbrace{\mathrm{R}(f_\theta(X); \mathcal{G})}_{\text{proposed regularizer}}, \tag{11}$$

with two positive parameters $\lambda_1, \lambda_2$. The first term is the standard UCE loss function. The second term is regarded by GPN as an entropy regularizer. The last term, R, is chosen to be either $\mathrm{R}_D$ or $\mathrm{R}_\alpha$, a decision implemented through hyperparameter tuning. We have included a theoretical result in Appendix B that provides a rationale for the proposed distance regularizations.

# 5 Experiments

In this section, we conduct extensive experiments on two tasks of OOD detection and misclassification detection. We compare the proposed framework (11) for uncertainty estimation of semi-supervised node classification using 8 datasets with a comparison to 5 baseline methods. The code is available at https://github.com/neoques/Graph-Posterior-Network.

## 5.1 Experiment Setup

**Datasets** We use three citation networks (i.e. CoraML, CiteSeer, Pubmed) [4], two co-purchase datasets [31] (i.e. AmazonComputers, AmazonPhotos), two coauthor datasets [31] (i.e. CoauthorCS and CoauthorPhysics) and a large dataset OGBN Arxiv [16]. A detailed description of these datasets is in Appendix E. We show the result of three citation datasets in the main paper and the remaining results in Appendix F.

**Baselines** We present the results for uncertainty estimation using five baseline methods. Among these, two evidence collection models, namely graph kernel density estimation (GKDE) [39] and label propagation (LP) [32], assuming that OOD nodes are located far away from the training nodes, while easily misclassified nodes reside near the boundaries between classes. We compare to a modified GCN model, referred to as VGCN-Energy [21], a Bayesian-based model, called GKDE-GCN [39], and GPN [32] as baselines in our evaluation. We also introduce a Graph Neural Network called APPNP [12] as one baseline for the misclassification detection task and report the ROC score. Details of these baselines can be found in Appendix D.

**Metrics** To assess the classification performance of ID nodes, we rely on the metric **ID-ACC**, which calculates the fraction of correct predictions among all predictions. As for evaluating uncertainty estimation, we employ the metrics **AUC-ROC** and **AUC-PR** as evaluation measures. The rankings are based on the scores of epistemic or aleatoric uncertainty. OOD detection is treated as a binary classification task, where the positive class corresponds to OOD nodes and the negative class pertains to ID nodes. Please refer to (2) for the calculation of aleatoric uncertainty and epistemic uncertainty. For the Dirichlet-based models, the epistemic uncertainty has a similar practical interpretation to vacuity in the belief theory, assessed using AUC scores. On the other hand, in VGCN-Energy, the calculation of epistemic uncertainty is based on the energy value and is represented as $u_i^{\text{epis}} = \text{energy}$. The misclassification detection task is also a binary classification problem, where the positive cases correspond to wrongly classified nodes and the negative cases represent correctly classified nodes. The calculation of uncertainty for misclassification detection is performed in the same manner as OOD detection except that $u_i^{\text{epis}} = -\max_k \alpha_i^k$. Prior studies [32, 39] has indicated that aleatoric uncertainty is generally more effective for identifying misclassifications, whereas epistemic uncertainty is more appropriate for detecting out-of-distribution instances.

**Model Setup** For all the baseline methods, we maintain consistency by employing the same set of model hyperparameters as provided by GPN. Specifically for some model hyperparameters such as latent dimension and weight decay, we adopt the same settings as GPN. Inspired by [26], we explore multiple choices of activation functions in the representation networks. In addition to the default ReLU used in GPN, we experiment with Sigmoid and GELU activation functions. Through empirical evaluation, we discover that the choice of activation function significantly impacts the performance of certain datasets, which is demonstrated in Appendix E.4. Besides, hyperparameters that we tune include entropy regularization weight, distance-based regularization format (whether $\text{R}_D$ or $\text{R}_\alpha$), and weighting parameters ($\lambda_1, \lambda_2$), which are optimized based on the validation cross-entropy for each specific dataset. For a comprehensive overview of the hyperparameter configuration and ablation study, please refer to Appendix D.

## 5.2 Results

**OOD Detection** OOD detection aims to detect whether an input example is OOD given the predicted uncertainty estimation. For the semi-supervised node classification, we adopt the **Left-Out-Classes** setting where we assume several categories as OOD (details can be found in Appendix D), as considered in [7, 39]. Different from the independent input setting, we retain the OOD nodes in the graph but exclude their labels from the training and validation sets. This implies that the loss function does not involve OOD labels, but the model has encountered the OOD node features during the training phase. Similarly to [32], we also remove the last graph propagation layer for comparison as "w/o network" where the final result only depends on the node features and no graph structure involved. This configuration, referred to as the "w/o network" setting, results in a final output that solely relies on the node features, with no involvement of the graph structure.

The results on CoraML, Citeseer, and PubMed are presented in Table 1 and the results on the other 5 datasets are shown in Table 6 in Appendix E. Our model achieves the best ID accuracy for four datasets and demonstrates comparable performance to GPN for the remaining four datasets Furthermore, we observe an improvement in the ROC (Receiver Operating Characteristic) rankings based on predicted epistemic uncertainty with propagation, ranging from +1% to +8% compared to GPN. Consistent with previous studies [32], our results demonstrate that prediction models incorporating evidence propagation consistently outperform those without propagation across all datasets. This observation highlights the significant impact of graph structure on uncertainty estimation. Moreover, when comparing aleatoric uncertainty and epistemic uncertainty as ranking scores, we find that epistemic

uncertainty outperforms aleatoric uncertainty in the OOD detection task. This finding aligns with literature [39, 32] and emphasizes the superiority of epistemic uncertainty for OOD detection.

Table 1: AUROC and AUPR for the OOD Detection

| Data | Model | ID-ACC | AUROC | | | AUPR | | |
|---|---|---|---|---|---|---|---|---|
| | | | Alea w/ | Epi w/ | Epi w/o | Alea w/ | Epi w/ | Epi w/o |
| CoraML | LP | 86.40 | 83.78 | 80.86 | n.a. | 74.80 | 71.15 | n.a. |
| | GKDE | 83.02 | 74.46 | 71.86 | n.a. | 66.19 | 64.05 | n.a. |
| | VGCN-Energy | 89.66 | 81.70 | 83.15 | n.a. | 75.67 | 78.44 | n.a. |
| | GKDE-GCN | 89.33 | 82.23 | 82.09 | n.a. | 75.88 | 77.03 | n.a. |
| | GPN | 88.51 | 83.25 | 86.28 | **80.95** | 75.79 | 79.97 | **72.81** |
| | Ours | **90.06** | **83.94** | **87.20** | 76.12 | **76.26** | **80.36** | 63.32 |
| Citeseer | LP | 57.34 | 65.99 | 67.54 | n.a. | 48.12 | 48.59 | n.a. |
| | GKDE | 49.62 | 63.75 | 63.91 | n.a. | **56.74** | 56.79 | n.a. |
| | VGCN-Energy | 70.79 | 72.16 | 76.08 | n.a. | 53.71 | 58.35 | n.a. |
| | GKDE-GCN | 70.76 | 73.34 | 76.19 | n.a. | 54.25 | **59.07** | n.a. |
| | GPN | 69.79 | 72.46 | 70.74 | 66.65 | 55.14 | 50.52 | 44.93 |
| | Ours | **72.51** | **75.22** | **78.98** | **73.21** | 62.30 | 58.63 | **52.73** |
| PubMed | LP | 89.18 | **80.32** | 79.64 | n.a. | **71.01** | 72.98 | n.a. |
| | GKDE | 88.16 | 69.66 | 68.47 | n.a. | 55.81 | 54.33 | n.a. |
| | VGCN-Energy | **94.77** | 72.58 | 72.63 | n.a. | 60.54 | 60.63 | n.a. |
| | GKDE-GCN | 94.66 | 73.53 | 74.47 | n.a. | 61.36 | 61.96 | n.a. |
| | GPN | 94.08 | 71.84 | 73.91 | 71.2 | 57.92 | 67.19 | 59.72 |
| | Ours | 93.84 | 75.23 | **81.76** | **77.79** | 60.75 | **78.16** | **69.19** |

Alea: Aleatoric, Epi.: Epistemic, w/: with propagation, w/o: without propagation

AUROC and AUPR for the OOD Detection: ID-ACC means the accuracy on ID nodes. AUROC and AUPR scores are given for OOD detection based on different uncertainty metrics, where [Alea w/] is the aleatoric score with propagation layer, [Epi w/] is the epistemic score with propagation layer and [Epi w/o] is the epistemic score without propagation. n.a. means either model or metric not applicable.

**Misclassification Detection**    In addition to OOD detection, we conduct misclassification detection on the clean graph for evaluating the predictive uncertainty estimation. Table 2 presents the results for three datasets, while Table 7 in Appendix D is for the other 5 datasets. We observe a significant improvement ranging from +12% to +50% in our model's AUC-PR scores. While it is true that our method performs worse than the best of the baselines in terms of AUROC, the differences are within approximately 3% for six of the eight datasets.: Amazon Computers, Amazon Photos, Coauthor CS, Coauthor Physics, and ODBG Arxiv, and PubMed. Despite having a lower AUROC compared to the best of the baselines, our method exhibits a higher AUPR. This suggests that our method may excel at identifying true positives among the top-ranked nodes when compared to GPN, while the best baselines may be more effective at distinguishing between true positives and negatives among the lower-ranked nodes.

Table 2: AUROC and AUPR for the Misclassification Detection

| Data | Model | AUROC | | AUPR | |
|---|---|---|---|---|---|
| | | Alea w/ | Epi w/ | Alea w/ | Epi w/ |
| CoraML | APPNP | **83.64** | n.a | 48.39 | n.a |
| | VGCN-Energy | 81.02 | n.a | 48.30 | n.a |
| | GKDE-GCN | 80.80 | 76.83 | 49.61 | 45.87 |
| | GPN | 81.19 | **78.10** | 49.51 | 44.42 |
| | Ours | 75.8 | 69.85 | **89.95** | **88.20** |
| CiteSeer | APPNP | 73.55 | n.a. | 51.70 | n.a. |
| | VGCN-Energy | 74.64 | n.a | 48.30 | n.a. |
| | GKDE-GCN | 75.45 | 73.83 | 54.78 | 53.57 |
| | GPN | **75.89** | **74.16** | 60.78 | 59.32 |
| | Ours | 69.15 | 68.62 | **72.67** | **72.36** |
| PubMed | APPNP | 80.98 | n.a. | 37.79 | n.a. |
| | VGCN-Energy | **81.16** | n.a | 38.24 | n.a |
| | GKDE-GCN | 80.95 | 73.99 | 39.64 | 33.19 |
| | GPN | 80.46 | **75.38** | 40.74 | 35.11 |
| | Ours | 80.13 | 72.87 | **95.41** | **92.79** |

Alea: Aleatoric, Epi.: Epistemic, w/: with propagation, w/o: without propagation

AUROC and AUPR for misclassification detection: AUROC and AUPR scores are given for misclassification detection based on different uncertainty metrics, where [Alea w/] is the aleatoric score with propagation layer, [Epi w/] is the epistemic score with propagation layer and [Epi w/o] is the epistemic score without propagation. n.a. means either model or metric not applicable

## 5.3   Ablation Study

Our proposed model differs from the GPN model in three main aspects. First, we use validation cross entropy (CE) instead of hold-out datasets to select hyperparameters. Second, we consider the

activation function for the MLP layers as one of the hyperparameters for selection. We expect feature value rescaling through non-linear activation of feature values to affect the density predictions. Third, we incorporate one of the proposed distance-based regularization terms to the loss function used in GPN.We conduct an ablation study to demonstrate the contribution of these three components. The results for CiteSeer and PubMed are shown in Table 3 and the remaining datasets are included in Appendix E. Hyperparameter tuning using validation cross-entropy improves GPN's performance, especially in cases where the choice of activation function has a significant impact on specific datasets. Additionally, we consistently observe performance enhancements from the distance-based regularization in both datasets, demonstrating the effectiveness of the proposed distance awareness regularization term.

Table 3: Ablation Study with OOD Detection task

| Data | Model | ID-ACC | AUROC | | | AUPR | | |
|------|-------|--------|-------|-----|-----|------|-----|-----|
| | | | Alea w/ | Epi w/ | Epi w/o | Alea w/ | Epi w/ | Epi w/o |
| Citeseer | GPN | 69.79 | 72.46 | 70.74 | 66.65 | 55.14 | 50.52 | 44.93 |
| | GPN-CE | 70.98 | 74.20 | 73.75 | 68.41 | 58.12 | 53.55 | 46.60 |
| | GPN-CE-ACT | 71.96 | 74.72 | 77.97 | 72.28 | 60.41 | 56.04 | 50.73 |
| | GPN-CE-GD | **72.51** | **75.22** | **78.98** | **73.21** | **62.30** | **58.63** | **52.73** |
| PubMed | GPN | **94.08** | 71.84 | 73.91 | 71.2 | 57.92 | 67.19 | 59.72 |
| | GPN-CE | 93.84 | 74.19 | 78.32 | 74.50 | 59.85 | 74.11 | 64.55 |
| | GPN-CE-ACT | 93.84 | 74.19 | 78.32 | 74.50 | 59.85 | 74.11 | 64.55 |
| | GPN-CE-GD | 93.84 | **75.23** | **81.76** | **77.79** | **60.75** | **78.16** | **69.19** |

Alea: Aleatoric, Epi.: Epistemic, w/: with propagation

GPN refers to the original GPN paper with its default hyperparameters and ReLU as the middle activation function, GPN-CE is the original GPN model with re-tuned Dirichlet entropy regularization weight based on validation cross-entropy; GPN-CE-ACT is the original GPN model with re-tuned entropy regularization weight and activation function based on cross-entropy; GPN-CE-GD/(Ours) adds the distance-based regularization term while tuning the two weights and activation function.

## 6 Limitations

Our theoretical analyses mainly study the limitations of the UCE loss function when separating OOD from ID nodes in the learned representation space. If we include the entropy-based regularization term in Equation (6) with a sufficiently large weight $\lambda$, some of our theoretical findings may not remain applicable. However, the entropy-based regularization term is designed to favor smooth Dirichlet distributions but not to preserve the distance between OOD and ID nodes. We conjecture that the resulting loss function is still insufficient to learn a representation space that separates OOD from ID nodes, even though they are separable in the original feature space. In addition, our proposed regularization terms in Section 4.2 are more effective for homophily graphs than heterophily graphs, as neighboring nodes are less likely to belong to the same class in a heterophily graph than those in a homophily graph.

## 7 Conclusion

This paper contributed to a better understanding of uncertainty quantification for node classification. We investigated the limitations of the widely used UCE loss function. Motivated by the theoretical analysis, we proposed a distance-based regularization that helps learn a representation network that is more effective for the uncertainty quantification task. Experimentally, we demonstrated our approach outperforms the state-of-the-art in two specific applications of uncertainty quantification for node classification: OOD detection and misclassification detection.

## 8 Acknowledgments

This work is supported by the National Science Foundation (NSF) under Grant No #2220574, #2107449, #1846690, and #1750911. The work of Feng Chen is also supported by the Intelligence Advanced Research Projects Activity (IARPA) via Department of Interior/Interior Business Center (DOI/IBC) contract number 140D0423C0026. The US Government is authorized to reproduce and distribute reprints for Governmental purposes notwithstanding any copyright annotation thereon. Disclaimer: The views and conclusions contained herein are those of the authors and should not be interpreted as necessarily representing the official policies or endorsements, either expressed or implied, of IARPA DOI/IBC or the U.S. Government.

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

The supplementary materials are organized as follows. Appendix **A** provides the background knowledge on the Dirichlet distribution. In Appendix **B** we review the architecture of the graph posterior network (GPN) [32] together with our discussion on oversight of [32, Theorem 1]. Appendix **C** details the proofs of all the theorems and corollaries discussed in the main paper. We provide detailed descriptions of baseline models, datasets, and hyperparameter tuning for the experiments in Appendix **D**. Lastly, Appendix **E** includes more experimental results that we are unable to fit into the paper.

## A    Dirichlet Distribution

A non-degenerate Dirichlet distribution, denoted by $\mathrm{Dir}(\boldsymbol{\alpha})$, is parameterized by the concentration parameters $\boldsymbol{\alpha} = [\alpha_1, \cdots, \alpha_K]^{\mathsf{T}}$ with $\alpha_k > 1$ for $k \in [K]$. More specifically, the Dirichlet distribution with parameters $\alpha_1, \cdots, \alpha_K$ has a probability density function (pdf) given by

$$\mathrm{pdf}(\mathbf{p}; \boldsymbol{\alpha}) = \frac{1}{B(\boldsymbol{\alpha})} \prod_{k=1}^{K} p_k^{\alpha_k - 1}, \tag{12}$$

where $\{p_k\}_{k=1}^K$ belongs to the standard probability simplex, thus $\sum_{k=1}^K p_k = 1$ and $p_k \in [0, 1], \forall k \in [K]$, and the normalizing constant $B(\boldsymbol{\alpha})$ is expressed in terms of the Gamma function $\Gamma(\cdot)$, i.e.,

$$B(\boldsymbol{\alpha}) = \frac{\prod_k \Gamma(\alpha_k)}{\Gamma(\Sigma_k \alpha_k)}. \tag{13}$$

Under the semi-supervised learning setting, a set of labels is available, denoted by $\mathbb{L} \subset \mathcal{V}$. For $i \in \mathbb{L}$, the class label $y_i \in \{1, \ldots, K\}$ can be converted into a one-hot vector $\mathbf{y}_i$ with $y_{ik} = 1$ if the sample belongs to the $k$-th class and $y_{ij} = 0$ for $j \neq k$. By arranging $\boldsymbol{\alpha}_i$ and $\mathbf{y}_i$ into matrices $\mathcal{A} := [\boldsymbol{\alpha}_i]_{i \in \mathbb{L}}$ and $Y := [\mathbf{y}_i]_{i \in \mathbb{L}}$, the UCE loss function is defined as:

$$\mathrm{UCE}(\mathcal{A}, Y) = \sum_{i \in \mathbb{L}} \sum_{k \in [K]} y_{ik}(\Psi(\alpha_{i0}) - \Psi(\alpha_{ik})), \tag{14}$$

where $\alpha_{i0} = \sum_{k=1}^K \alpha_{ik}$ and $\Psi$ is the digamma function given by

$$\Psi(x) = \frac{\Gamma'(x)}{\Gamma(x)}.$$

## B    Detailed Framework of GPN

In this section, we review the architecture of GPN, followed by two examples that reveal a limitation of Theorem 1 in the GPN paper [32].

**Multi-layer Perceptron (MLP).**   Instead of deep convolution layers used in many neural networks designed for image classification task [7], GPN utilizes two simple perceptron layers with ReLU activation function as the encoding network, which maps high dimensional data to a latent space with a much smaller dimension, avoiding the curse of dimensionality for the density estimation on a (mapped) latent representation [25]. As each node is independent of the others in this step, the encoding map only considers the node features without any graph structure involved. Mathematically, the mapping can be expressed by

$$\boldsymbol{z}_i = f(\boldsymbol{x}_i; \boldsymbol{\theta}) = W_2 \sigma(W_1 \boldsymbol{x}_i + \mathbf{1}^{\mathsf{T}} \mathbf{b}_1) + \mathbf{1}^{\mathsf{T}} \mathbf{b}_2,$$

where $\theta := \{W_1, W_2, \boldsymbol{b}_1, \boldsymbol{b}_2\}$ denotes a set of learning parameters. For simplicity we use the notation $\mathbf{z}_i = f_\theta(\mathbf{x}_i)$.

**Normalizing Flow.**   Normalizing flow is used to estimate the density $\mathbb{P}(\boldsymbol{z}_i | k; \boldsymbol{\phi})$ for $k \in [K]$ and learning parameters $\phi$ as an invertible transformation $q(\cdot; k)$ of a base distribution, e.g. Normal distribution, which denotes the distribution of class $k$ in the latent space. The default flow in GPN is the radial flow [27], given by

$$q(\mathbf{z}; k) = \mathbf{z} + \frac{\beta(\mathbf{z} - \mathbf{z}_0)}{\gamma + \|\mathbf{z} - \mathbf{z}_0\|}$$

$$\mathbb{P}(\mathbf{z}_i | k; \phi) = p_z(q^{-1}(\mathbf{z}_i; k)) |\det \frac{\partial q^{-1}(\cdot; k)}{\partial \mathbf{z}}|.$$

where $\mathbf{z}_0$ is a reference point and $p_z(\cdot) \sim \mathcal{N}(0, 1)$. After estimating the density of the node $i$ belonging to a specific class $k$, the pseudo evidence counts are scaled to the probability, i.e., $\beta_i^k \propto \mathbb{P}(\boldsymbol{z}_i | k; \boldsymbol{\phi})$, GPN sets

$$\beta_i^k := g_\phi(\boldsymbol{z}_i)_k = N_k \cdot \mathbb{P}(\boldsymbol{z}_i | k; \boldsymbol{\phi}),$$

where $N_k$ is the number of training nodes that belong to the class $k$.

**Personalized Page Rank.** GPN applies a personalized page rank (PPR) module to diffuse the evidence among neighboring nodes. It is motivated by the work of Approximate Personalized Propagation of Neural Predictions (APPNP) [12] that is designed to decouple the prediction (only based on node features) with any encoding network and propagate with a personalized page rank (PPR) module (only based on edge information). In particular, PPR provides a personalized influence score matrix for each node that considers $L$ hop of neighbors without involving any new parameters to learn and $L$ is a hyperparameter:

$$\boldsymbol{\beta}^{(l+1)} = (1-\gamma)\hat{A}\boldsymbol{\beta}^{(l)} + \gamma\boldsymbol{\beta}^{(0)},$$

where $\gamma$ is a hyper-parameter relating to the teleport probability, $\hat{A}$ denotes the symmetrically normalized graph adjacency matrix with added self-loops (i.e., $\hat{A} := D^{-1/2}AD^{-1/2}$ with the standard adjacency matrix $A$), and $l$ denotes the layer index with $\boldsymbol{\beta}^{(0)}$ obtained after the normalizing flow. The output of PPR is a set of concentration parameters, denoted by $\boldsymbol{\alpha} = h_\gamma(\boldsymbol{\beta}^{(0)})$.

Collectively for MLP, normalizing flow, and PPR, the network in GPN can be expressed by

$$\boldsymbol{\alpha}_i = 1 + h_\gamma(g_\phi(f_\theta(\mathbf{x}_i))), \tag{15}$$

for each node $i$, where the addition of 1 guarantees that the concentration parameter is strictly positive. In addition, an entropy regularization was considered by GPN defined by,

$$\mathbb{H}(\text{Dir}(\boldsymbol{\alpha}_i)) = \log B(\boldsymbol{\alpha}_i) + (\alpha_{i0} - K)\Psi(\alpha_{i0}) - \sum_{k=1}^{K}(\alpha_{ik} - 1)\Psi(\alpha_{ik}), \tag{16}$$

where $\alpha_{i0} = \sum_{k=1}^{K} \alpha_{ik}$.

Next, we provide two examples to describe oversight of [6, Theorem 1] and [32, Theorem 1] in the sense that both theorems assume an impossibility. Particularly the assumption is that a two-layer ReLU network can be represented by a set of affine mappings, each being full rank, from a finite set of regions to the latent space. However, we construct Example 9 and Example 10 to show this assumption is impossible.

**Example 9.** *We start with a simple case where a two-layer ReLU network with input, hidden layer, and output of a scalar (1-dimensional) is considered for an easier interpretation of the results. One simple example of a two-layer ReLU network is expressed by*

$$z = f_\theta(x) = 1 \cdot \sigma_{ReLU}(1 \cdot x + 0) + 0. \tag{17}$$

*Following [15], we split the latent space into two affine regions, i.e.,*

$$z = \begin{cases} x & \text{if } x \in [0, \infty) \\ 0 & \text{if } x \in (-\infty, 0], \end{cases} \tag{18}$$

*labeled by $Q^{(0)} = [0, \infty)$ and $Q^{(1)} = (-\infty, 0]$. We see the associated $V^{(0)} = 1$ and $V^{(1)} = 0$ in the affine representation (17) that certainly do not have independent rows, as required by [6, Theorem 1].*

Example 10 extends the 1D case in Example 9 into a higher $d$-dimension, showing that there is always at least one affine region that produces a single value, i.e. $f_\theta(Q^{(l)^*}) = \{\mathbf{v}\}$ when mapped into a ReLU network $f_\theta$.

**Example 10.** *We consider the ReLU network,*

$$f_\theta(\mathbf{x}) = C\sigma_{ReLU}(B\mathbf{x}), \tag{19}$$

*where $B, C \in \mathbb{R}^{d \times d}$ are matrices of full rank. Denote $\mathbf{x}_j$ to be the solution to the equation,*

$$-\mathbf{e}_j = B\mathbf{x}, \tag{20}$$

*where $\mathbf{e}_j$ is the jth Euclidean standard basis. As $B$ is assumed to be full rank, there is the unique solution of the corresponding $\mathbf{x}_j$.*

*Notice that the polytope,*

$$S = \left\{ \sum_{j=1}^{d} a_j \mathbf{x}_j \,\middle|\, a_j \geq 0 \right\}, \tag{21}$$

*has non-zero measure in $\mathbb{R}^d$. Note that the ReLU network is constant by construction, as $\sigma_{ReLU}(-\mathbf{e}_j) = \mathbf{0}$. In other words, we have for $x \in S$ that*

$$\mathbf{z} = f_\theta(\mathbf{x}) = C\sigma_{ReLU}(B\mathbf{x}) = C\mathbf{0} = \mathbf{0}. \tag{22}$$

*As in the previous example, the existence of $S$ means that there exists some $V^{(\cdot)} = \mathbb{0}_{d,d}$ which contradicts the assumption that all $V$s have independent rows. Under this setting, the density does not approach zero, which is the conclusion of Theorem 1 in [32].*

## C Proofs

In this section, we provide the proof of all the theorems and corollaries. Note that until Theorem 8, We ignore the graph component $h_\gamma$ and focus solely on the representational layer $f_\theta$ and normalizing flow layer $g_\phi$.

*Proof of Theorem 1 and Corollary 3.* As $f_\theta$ is arbitrary by assumption, we choose it in such a way that it maps a point in $\mathcal{X}_k$ to a point inside the ball centered at $\mathbf{z}_k$ with radius $r_k$, denoted by $B(\mathbf{z}_k, r_k)$,

$$f_\theta : \mathcal{X}_k \to \mathcal{Z}_k \subset B(\mathbf{z}_k, r_k), \tag{23}$$

where $\mathbf{z}_k \in \mathcal{Z}$ with a minimal distance $R$ between any two of them, i.e., $d(\mathbf{z}_k, \mathbf{z}_m) > R, \forall k, m \in [K]$, and we define $r > r_k, \forall k \in [K]$. We then choose the normalizing flow to be,

$$g_\phi(\mathbf{z}; k) = 1 + N_k \cdot \begin{cases} \frac{1}{\mathrm{Vol}(B(\mathbf{z}_k, r_k))}, & \text{if } \mathbf{z} \in B(\mathbf{z}_k, r_k), \\ 0, & \text{otherwise.} \end{cases} \tag{24}$$

We add the value of 1 in the normalizing flow to produce valid evidence measures. We also assume that $N_k = \mu(\mathcal{X}_k) > 0$, where $\mu$ is the Lebesgue measure function.

The global minimum of UCE occurs when UCE is equal to 0 for every class. Recall that

$$\sum_{k \in [K]} \mathrm{UCE}\,(g(\mathcal{Z}_k), Y) = \sum_{k \in [K]} \int_{\mathcal{Z}_k} \left( \Psi \left( \sum_{m \in [K]} g_m(\mathbf{z}) \right) - \Psi(g_k(\mathbf{z})) \right) d\mu. \tag{25}$$

Using (23), we consider an upper bound of the right-hand side by integrating over the larger region, that is,

$$\sum_{k \in [K]} \int_{B(\mathbf{z}_k, r_k)} \left( \Psi \left( \sum_{m \in [K]} g_m(\mathbf{z}) \right) - \Psi(g_k(\mathbf{z})) \right) d\mu, \tag{26}$$

$$= \sum_{k \in [K]} \mathrm{Vol}(B(\mathbf{z}_k, r_k)) \cdot \left( \Psi \left( K + \frac{N_k}{\mathrm{Vol}(B(\mathbf{z}_k, r_k))} \right) - \Psi \left( 1 + \frac{N_k}{\mathrm{Vol}(B(\mathbf{z}_k, r_k))} \right) \right). \tag{27}$$

According the recurrence relation of the digamma function: $\Psi(x + 1) = \Psi(x) + 1/x$, we readily derive that,

$$\sum_{k \in [K]} \mathrm{UCE}\,(g(\mathcal{Z}_k), Y) \leq \sum_{k \in [K]} \mathrm{Vol}(B(\mathbf{z}_k, r_k)) \cdot \sum_{m=1}^{K-1} \left( K - m + \frac{N_k}{\mathrm{Vol}(B(\mathbf{z}_k, r_k))} \right)^{-1}. \tag{28}$$

Taking the limit of the right-hand side yields

$$\lim_{r \to 0} \sum_{k \in [K]} \mathrm{Vol}(B(\mathbf{z}_k, r_k)) \cdot \sum_{m=1}^{K-1} \left( K - m + \frac{N_k}{\mathrm{Vol}(B(\mathbf{z}_k, r_k))} \right)^{-1}, \tag{29}$$

$$= \sum_{k \in [K]} \lim_{r_k \to 0} \mathrm{Vol}(B(\mathbf{z}_k, r_k)) \cdot \lim_{r_k \to 0} \sum_{m=1}^{K-1} \left( K - m + \frac{N_k}{\mathrm{Vol}(B(\mathbf{z}_k, r_k))} \right)^{-1}. \tag{30}$$

It is straightforward for the following two limits to hold,

$$\lim_{r_k \to 0} \mathrm{Vol}(B(\mathbf{z}_k, r_k)) = 0, \tag{31}$$

$$\lim_{r_k \to 0} \sum_{m=1}^{K-1} \left( K - m + \frac{N_k}{\mathrm{Vol}(B(\mathbf{z}_k, r_k))} \right)^{-1} = 0, \tag{32}$$

thus leading to

$$\lim_{r \to 0} \sum_{k \in [K]} \mathrm{Vol}(B(\mathbf{z}_k, r_k)) \cdot \sum_{m=1}^{K-1} \left( K - m + \frac{N_k}{\mathrm{Vol}(B(\mathbf{z}_k, r_k))} \right)^{-1} = 0. \tag{33}$$

On the other hand, as $\Psi(\sum_{k \in [K]} g_k(\mathbf{z})) - \Psi(g_k(\mathbf{z})) \geq 0$, we have

$$0 \leq \sum_{k \in [K]} \int_{\mathcal{Z}_k} (\Psi \left( \sum_{m \in [K]} g_m(\mathbf{z}) \right) - \Psi(g_k(\mathbf{z})) d\mu = \sum_{k \in [K]} \mathrm{UCE}\,(g(\mathcal{Z}_k), Y), \tag{34}$$

which implies that UCE $\to 0$ as $r \to 0$. For $r = 0$, UCE is equal to zero, which leads to Corollary 3. $\qquad\square$

*Proof of Theorem 4.* We denote the parameters of $g_\phi$ that represent the true analytic solution $\hat{\phi} = \phi(\theta)$. Note that this is a function with respect to the choice of $\theta$, that is, the true distribution is dependent on the representational mapping. In this proof, we focus on finding a value of $\theta$ s.t.,

$$\hat{\theta} = \arg\min_{\theta} \text{UCE}(\alpha(\theta, \hat{\phi}), Y) = \arg\min_{\theta} \text{UCE}(\alpha(\theta, \phi(\theta)), Y). \tag{35}$$

As the true distribution is dependent on the representational mapping, we should consider a joint minimization problem with respect to both the representation map and density distribution.

We will separate the proof into two cases. Specifically, we prove Case 1 by contradiction, showing that if the set $\mathcal{Z}_k$ mapped to by $f_\theta$ from $\mathcal{X}_k$ has a non-zero measure, then the global minimizer $\text{UCE} = 0$ can not be achieved. We then prove Case 2, under the assumption that a true analytical solution may achieve density evidence at a point, by showing that we may achieve the global minimizer on a point set.

**Case 1: Non-Zero Measure.** Suppose the true distribution on this set is a non-degenerate distribution. As the natural definitions of a probability distribution $1 = \int_{\mathcal{Z}_k} d\mu$, the UCE loss can be expressed by

$$\sum_{k \in [K]} \text{UCE}\left(g(\mathcal{Z}_k), Y\right) = \sum_{k \in [K]} \int_{\mathcal{Z}_k} \left( \Psi\left( \sum_{m \in [K]} g_m(\mathbf{z}) \right) - \Psi(g_k(\mathbf{z})) \right) d\mu(\mathbf{z}). \tag{36}$$

In order for the measure of $\mathcal{Z}_k$ to have a density of $\epsilon > 0$, there exists a subset of $\mathcal{Z}_k$ with non-zero measure $\delta_k > 0$, denoted $\mathcal{Z}_k^*$. Using similar techniques as the proof of Theorem 1 in reverse, we obtain,

$$\sum_{k \in [K]} \text{UCE}\left(g(\mathcal{Z}_k), Y\right) \geq \sum_{k \in [K]} \int_{\mathcal{Z}_k} \left( \Psi\left(K + N_k\epsilon\right) - \Psi(1 + N_k\epsilon) \right) d\mu,$$

then for some $k \in [K]$ there exists some $\mathcal{Z}_k^*$,

$$\sum_{k \in [K]} \text{UCE}\left(g(\mathcal{Z}_k), Y\right) \geq \int_{\mathcal{Z}_k^*} \left( \Psi\left(K + N_k\epsilon\right) - \Psi(1 + N_k\epsilon) \right) d\mu,$$

$$= \delta_1 \cdot \left( \Psi\left(K + N_k\epsilon\right) - \Psi(1 + N_k\epsilon) \right).$$

As $\Psi$ is strictly increasing, then $\delta_2 = \Psi\left(K + N_k\epsilon\right) - \Psi(1 + N_k\epsilon) > 0$, which implies that

$$\sum_{k \in [K]} \text{UCE}\left(g(\mathcal{Z}_k), Y\right) \geq \sum_{k \in [K]} \delta_1 \cdot \delta_2 > 0.$$

Therefore, we prove that if $f_\theta$ maps to a measurable set, the UCE loss is necessarily non-zero.

**Case 2: Zero Measure Sets.** Corollary 3 shows that the zero UCE is achievable. If Case 1 fails, then we can conclude that only on a disjoint set $\mathcal{Z}_k$ with measure 0 for each $k$ is permissible to achieve the UCE to be 0. The exact choice of this set depends on the precise definitions of the probability distributions on a point set and their ability to achieve infinite densities. Here we constrain these possibilities by requiring the range of $f_\theta$ to have non-zero measure or to be a point set if having zero measure[2]. $\qquad\square$

*Proof of Theorem 6.* Pick $\mathbf{x} \in \mathcal{X}$ s.t. $d(\mathbf{x}, \mathbf{x}_k) > \delta$ for $x_k \in \mathcal{X}_k$ see that for any $f_\theta$ where $\theta \in \Gamma$ we have that $\mathbf{x}$ is necessarily not mapped to $\mathbf{z}_k \in \mathcal{Z}$ (if it were mapped in $\mathcal{Z}$ then the preimage would contain it and thus we would have $d(\mathbf{x}, \mathbf{x}_k) < \delta$, which is a contradiction with our selection of $x$). Recall that the density for any point mapped to $z_k$ is infinite. That is the density of the associated points mapped to the point set is necessarily infinite and the density of points mapped elsewhere is necessarily smaller, namely 0, with the associated evidence 1. Our selected $\mathbf{x}$ then has no evidence in favor of it belonging to a class $k$ while any point in $\mathbf{x}_k \in \mathcal{X}_k$ must have infinite evidence by our choice of a well-fit $\theta$. $\qquad\square$

*Proof of Corollary 7.* Notice that we can choose two types of such $\theta$,

**Case 1** Let $\theta$ for class $k$ be chosen such that,

$$f(\mathbf{x}) = \begin{cases} \mathbf{z}_k \text{ if } \mathbf{x} \in \mathcal{X}_k, \\ 0 \text{ otherwise.} \end{cases} \tag{37}$$

---

[2]We choose that the cardinality of the zero-measure set $f_\theta(\mathcal{X}_k)$ to be finite (rather than countably infinite) as we do not want to detail precise topological arguments (like compactness and boundedness) about the pointsets and their respective preimages.

**Case 2**

$$f(x) = \begin{cases} \mathbf{z}_k \text{ if } d(\mathbf{x}, \hat{\mathbf{x}}) < \delta \text{ for any } \hat{x} \in \mathcal{X}_k, \\ 0 \text{ otherwise.} \end{cases} \tag{38}$$

If $\mathbf{x} \in \mathcal{X}_k$ is mapped to $\mathbf{z}_k$ then it is endowed with infinite density, moreover, it is believed to be an ID node belonging to class $k$. Thus, the nearby OOD being detected for these UCE minimizers is determined by arbitrary choice. □

*Proof of Theorem 8.* First note that the ID nodes are mapped to have infinite evidence achieved at the points in the latent space $\mathcal{Z}_k$. As the representations of the OOD nodes are in $\mathcal{Z}_k$ they are also endowed with infinite evidence. That is graph layers can only help separate nodes by pulling them towards the center of their own classes w.r.t. to the representation space this is only helpful if their representations are separate to begin with.

□

Lastly, we give a toy example showing heuristically that the proposed regularization yields a better separation of the OOD nodes from IDs, compared to the original GPN model without the distance-based regularization.

**Example 11.** *Consider two ID classes (Class 1 and Class 2) and one OOD class with the following construction:*

1. *All nodes belonging to Class 1 have feature values sampled from $\mathbf{x}^{(1)} = [1, 0, 0]$.*

2. *All nodes belonging to Class 2 have feature values sampled from $\mathbf{x}^{(2)} = [-1, 0, 0]$.*

3. *All nodes belonging to the OOD class 2 have feature values sampled from $\mathbf{x}^{(OOD)} = [0, 1, v]$.*

4. *We sample $v$ from the uniform distribution $U(-1, 1)$ independently for each sample in each class.*

5. *All nodes are connected to every node within their own class, leading to a graph of homophily 1.*

6. *Suppose the density function is true density distribution*

7. *Denote the PPR layer by $\hat{h}$ that uses the right normalized adjacency matrix $AD^{-1}$ rather than symmetrically normalized $D^{-1/2}AD^{-1/2}$, used in APPNP.*

8. *Suppose $f_\theta$ is a linear function (i.e. no activation function) explicitly, $W = \begin{bmatrix} W_{11} & W_{12} \\ W_{21} & W_{22} \\ W_{31} & W_{32} \end{bmatrix}$.*

*Then GPN with our regularization can learn an embedding that makes it possible to separate classes 1, 2, and OOD nodes. Without regularization, OOD nodes lie between ID classes in the latent space.*

*Proof.* A simple calculation for the project leads to

$$\mathbf{z}_i = [XW]_i = \begin{cases} [W_{11}, W_{12}] & \text{for class 1 nodes} \\ [-W_{11}, -W_{12}] & \text{for class 2 nodes} \\ [W_{21} + vW_{31}, W_{22} + vW_{31}] & \text{for OOD nodes.} \end{cases} \tag{39}$$

Clearly, the values of $W_{31}$ and $W_{32}$ would be smaller with the distance minimization term applied than without, as $v$ is selected randomly. Moreover neither $W_{31}$ nor $W_{32}$ affects the model's ability to separate the two classes as desired. We explicitly calculate both UCE and the distance-based regularization in the objective function, while ignoring the Dirichlet regularization, thus leading to the following objective function,

$$L(Z, \alpha, Y; \mathcal{G}) = \text{UCE}(\hat{h}(g_\phi(f_\theta(x))), Y) + R(Z; \mathcal{G}). \tag{40}$$

First, we explicitly work out the distance-based regularization term

$$\begin{aligned} \text{R}(Z; \mathcal{G}) &= \sum_{(i,j) \in \mathcal{E}} \|\mathbf{z}_i - \mathbf{z}_j\|^2 \\ &= \sum_{(i,j) \in \mathcal{E}_1} \|[W_{11}, W_{12}] - [W_{11}, W_{12}]\|^2 + \sum_{(i,j) \in \mathcal{E}_2} \|[-W_{11}, -W_{12}] - [-W_{1,1}, -W_{12}]\|^2 \\ &\quad + \sum_{(i,j) \in \mathcal{E}_{\text{OOD}}} \left\| [W_{21} + v^{(i)}W_{31}, W_{22} + v^{(i)}W_{32}] - [W_{21} + v^{(j)}W_{31}, W_{22} + v^{(j)}W_{32}] \right\|^2 \\ &= \sum_{(i,j) \in \mathcal{E}_{\text{OOD}}} \left\| (v^{(i)} - v^{(j)})[W_{31}, W_{32}] \right\|^2, \end{aligned}$$

which is minimized when $W_{31}, W_{32}$ go to zero.

Next, we consider the UCE loss portion. See that as we estimate the true density using $g$ we will have no overlap between the two distributions $\mathcal{Z}_1, \mathcal{Z}_2$. We are left with $W = \begin{bmatrix} W_{11} & W_{12} \\ W_{21} & W_{22} \\ 0 & 0 \end{bmatrix}$,

$$Z_i = [XW]_i = \begin{cases} [W_{11}, W_{12}] \text{ for class 1 nodes} \\ [-W_{11}, -W_{12}] \text{ for class 2 nodes} \\ [W_{21}, W_{22}] \text{ for OOD nodes,} \end{cases} \tag{41}$$

If $W$ is to remain full rank this will necessarily require either $W_{21}$ or $W_{22}$ to be non-zero. Thus OOD nodes will be mapped as we see in (41) to some distinct values - which can be separated after the application of APPNP as we expect APPNP to only average the values within each class. □

# D   Additional Experimental Details

## D.1   Descriptions of Baselines

**Graph-based Kernel Dirichlet distribution Estimation (GKDE)** [39]: Based on the high homophily property of most graphs (neighboring nodes tend to share the same class label), GKDE derives the evidence with the help of the node-level distances (shortest path in the graph) with training nodes belonging to the same class.

**Label Propagation (LP)** [32]: Following the idea of GKDE, LP collects the evidence by relying on the density of labeled nodes in neighborhoods rather than distance. An initial condition per class is defined and then a Personalized Page Rank is used as the diffusion.

**VGCN-Energy** [21]: It is a GCN-based model with energy score as the uncertainty estimation which maps each node to a single, non-probabilistic scalar called the energy. The energy score can be calculated as follows

$$s_{\text{energy}}^i = -T \log \sum_{k=1}^{K} \exp^{\frac{l_i^k}{T}},$$

where $l$ is the predicted logits of a neural network and temperature parameter $T = 1$.

**GKDE-GCN** [39]: GKDE-GCN utilizes a GCN network to estimate the multisource uncertainty by a Dirichlet distribution and then sample probability as well as the class prediction. The evidence derived from the aforementioned GKDE is as a teacher of concentration parameters of Dirichlet Distribution, and another deterministic GCN predicting the probability is used as a teacher for sampled probability. The overall loss is composed of the KL divergence between these two teachers with the corresponding distribution and Bayes risk with respect to the squared loss of sampled class prediction.

**APPNP** [12]: Given that message passing neural network suffers from the over-smoothing problem that limits the depth of the neural network, APPNP proposed to decouple the prediction and propagation where the prediction depends on the node features and propagation depends on interactions between nodes through edges. APPNP first uses any kind of neural network to embed the input space and diffuses information with a personalized page rank. For large graphs, they use power iteration to approximate a topic-sensitive page rank.

**GPN** [32]: GPN applies a normalizing flow to estimate the density of each class in the latent space embedded with an encoding network and then propagates the scaled density as the evidence.

## D.2   Description of Datasets

We use three citation networks, labelled by CoraML, CiteSeer, Pubmed [4], two co-purchase Amazon datasets [31], labeled by Computers and Photos, two coauthor datasets [31], labeled by CoauthorCS and Physics, and a large dataset OGBN Arxiv [16]. We use the same train/val/test split of 5/15/80 as [32]. The details of the graphs and setups for the OOD detection are provided in Table 4.

Table 4: Dataset Description

|  | CoraML | CiteSeer | PubMed | Computers | Photos | Coauthor CS | Coauthor Physics | OGBN-Arxiv |
|---|---|---|---|---|---|---|---|---|
| #nodes | 2,995 | 4,230 | 19,717 | 13,752 | 7,650 | 18,333 | 34,493 | 169,343 |
| #edges | 16,316 | 10,674 | 88,648 | 491,722 | 238,162 | 163,788 | 495,924 | 2,315,598 |
| #features | 2879 | 602 | 500 | 767 | 745 | 6,805 | 8,415 | 128 |
| #classes | 7 | 6 | 3 | 10 | 8 | 15 | 5 | 40 |
| # left-out-classes | 3 | 2 | 1 | 5 | 3 | 4 | 2 | 15 |

## D.3 Hyper-parameter tuning

We follow the same setting with [32]. In detail, we use the Adam optimizer with a learning rate of 0.01. For VGCN-Energy, we use a temperature of $T = 1.0$. We carefully tune three hyperparameters: the distance-based regularization weight, Dirichlet entropy weight, and activation functions. We select the best parameters for each dataset separately that returns the highest validation cross-entropy. The detailed hyperparameters configuration is as Table 5.

Table 5: Hyperparameter configurations of proposed model

|  | Dirichlet Entropy Reg. Weight | Graph Distance Reg. Weight | Activation function |
|---|---|---|---|
| CoraML | 0 | $10^{-4}$ | GELU |
| CiteSeer | $10^{-4}$ | $10^{-9.5}$ | LogSigmoid |
| PubMed | $10^{-5}$ | $10^{-4}$ | RELU |
| Computers | $10^{-5}$ | $10^{-4}$ | RELU |
| Photos | $10^{-5}$ | $10^{-11}$ | RELU |
| Coauthor CS | 0 | $10^{-6}$ | RELU |
| Coauthor Physics | $10^{-4}$ | $10^{-4.5}$ | LogSigmoid |
| OGBN-Arxiv | $10^{-5}$ | $10^{-8}$ | RELU |

We also consider the following activation functions in the encoding network with element-wise operations,

$$\sigma_{\text{RELU}}(x) = \max(0, x),$$

$$\sigma_{\text{LogSigmoid}}(x) = \log\left((1 + \exp(-x))^{-1}\right),$$

$$\sigma_{\text{GeLU}}(x) = x\text{CDF}_{\mathcal{N}}(x)$$

$$\sigma_{\text{HardTanh}}(x) = \begin{cases} -1, x < -1 \\ x, -1 \leq x \leq 1 \\ 1, x > 1 \end{cases}.$$

ReLU is the most popular activation function used in the hidden layer of neural networks, which brings efficient computation by only activating neurons with positive outputs. Sigmoid is popularly used for probability prediction because its output is always in the range (0,1) with a smooth gradient. GeLU has better nonlinearity and is widely used in Natural Language processing and computer vision. HardTanh is a more computation-efficient version of Tanh.

# E   Additional Experiments

## E.1   Additional Experiments - OOD Detection

For Amazon Photos, Amazon Computers, Coauthor CS, Coauthor Physics, and OGBN Arxiv dataset, the OOD Detection results are shown in Table 6.

## E.2   Additional Experiments - Misclassification Detection

For Amazon Photos, Amazon Computers, Coauthor CS, Coauthor Physics, and OGBN Arxiv dataset, the Misclassification Detection results are shown in Table 7.

## E.3   Graph Distance Minimization

We plot the tSNE visualization of latent space with different distance-based regularization weights and symbol sizes denote the total evidence. We plot for coraML in Figure 2, CiteSeer in Figure 3, Coauthor CS in Figure 4, Coauthor Physics in Figure 5. With increasing weight, it tends to have a more separable latent representation for different categories while degenerate mappings occur when distance minimization is too large.

Table 6: OOD Detection (Cont.)

| Data | Model | ID-ACC | AUROC | | | AUPR | | |
|---|---|---|---|---|---|---|---|---|
| | | | Alea w/ | Epi w/ | Epi w/o | Alea w/ | Epi w/ | Epi w/o |
| Amazon Computers | LP | 83.28 | **86.74** | 83.88 | n.a. | **67.10** | 63.08 | n.a. |
| | GKDE | 71.41 | 75.14 | 73.58 | n.a. | 49.21 | 47.68 | n.a. |
| | VGCN-Energy | 88.95 | 82.76 | 83.43 | n.a. | 57.49 | 60.64 | n.a. |
| | GKDE-GCN | 82.73 | 77.03 | 70.32 | n.a | 49.81 | 45.92 | n.a |
| | GPN | 88.48 | 82.49 | 87.63 | **74.55** | 56.78 | 67.94 | **48.03** |
| | Ours | 89.88 | 83.56 | 89.26 | 71.82 | 58.51 | 71.06 | 43.35 |
| Amazon Photos | LP | 89.27 | **94.24** | 90.26 | n.a. | **90.24** | 85.55 | n.a. |
| | GKDE | 85.94 | 76.51 | 60.83 | n.a. | 66.72 | 59.09 | n.a. |
| | VGCN-Energy | 94.24 | 82.44 | 79.64 | n.a. | 72.60 | 71.71 | n.a. |
| | GKDE-GCN | 89.84 | 73.65 | 69.09 | n.a | 62.45 | 59.68 | n.a |
| | GPN | 94.10 | 82.72 | 91.98 | 76.57 | 74.55 | 86.29 | 64.00 |
| | Ours | 94.40 | 83.51 | 92.30 | 78.10 | 77.65 | 87.36 | 65.39 |
| Coauthor CS | LP | 86.40 | 83.78 | 80.86 | n.a. | 74.8 | 71.15 | n.a |
| | GKDE | 78.84 | 79.32 | 77.59 | n.a. | 66.30 | 64.69 | n.a. |
| | VGCN-Energy | 93.07 | **85.35** | 87.33 | n.a. | **80.87** | 82.79 | n.a. |
| | GKDE-GCN | **93.13** | 85.02 | 84.45 | n.a. | 80.15 | 77.90 | n.a. |
| | GPN | 88.21 | 69.49 | **92.90** | 88.84 | 55.41 | 90.28 | 86.54 |
| | Ours | 89.24 | 70.12 | 92.37 | **91.38** | 56.20 | **91.17** | 90.45 |
| Coauthor Physics | LP | 95.39 | **91.78** | 90.03 | n.a. | 70.58 | 69.63 | n.a. |
| | GKDE | 93.30 | 87.02 | 84.64 | n.a. | 57.00 | 52.49 | n.a. |
| | VGCN-Energy | **97.96** | 90.29 | 91.08 | n.a. | 63.63 | 69.41 | n.a. |
| | GKDE-GCN | 97.95 | 87.38 | 84.62 | n.a. | 57.97 | 56.30 | n.a. |
| | GPN | 97.40 | 85.20 | **94.51** | 89.63 | 61.89 | **83.73** | 66.44 |
| | Ours | 97.44 | 85.28 | 94.42 | **90.36** | 62.80 | 83.61 | **70.62** |
| OGBN Arxiv | LP | 66.84 | **80.04** | **75.22** | n.a. | **65.21** | 67.69 | n.a. |
| | GKDE | 51.51 | 68.12 | 65.80 | n.a. | 47.22 | 45.23 | n.a. |
| | VGCN-Energy | **75.61** | 64.91 | 64.50 | n.a. | 42.72 | 42.41 | n.a |
| | GKDE-GCN | 73.89 | 68.84 | 72.44 | n.a. | 49.71 | 52.23 | n.a. |
| | GPN | 73.84 | 66.33 | 74.82 | 62.17 | 46.35 | 58.71 | **43.01** |
| | Ours | 71.30 | 66.98 | 74.52 | **62.75** | 47.48 | 56.97 | 41.48 |

Alea: Aleatoric, Epi.: Epistemic, w/: with propagation, w/o: without propagation

Table 7: AUROC and AUPR for the Misclassification Detection (Cont.)

| Data | Model | AUROC | | AUPR | |
|---|---|---|---|---|---|
| | | Alea w/ | Epi w/ | Alea w/ | Epi w/ |
| Amazon Computers | APPNP | 79.75 | n.a. | 45.10 | n.a. |
| | VGCN-Energy | 82.08 | n.a. | 45.53 | n.a. |
| | GKDE-GCN | 79.66 | 73.66 | 63.26 | 56.93 |
| | GPN | **82.20** | **77.58** | 47.93 | 41.80 |
| | Ours | 80.75 | 74.87 | **93.12** | **90.11** |
| Amazon Photos | APPNP | 85.74 | n.a. | 37.00 | n.a. |
| | VGCN-Energy | **87.94** | n.a. | 48.35 | n.a. |
| | GKDE-GCN | 84.11 | 75.07 | 54.35 | 45.43 |
| | GPN | 87.21 | **83.38** | 46.32 | 37.07 |
| | Ours | 84.42 | 81.61 | **96.89** | **96.70** |
| Coauthor CS | APPNP | **89.92** | n.a. | 37.98 | n.a. |
| | VGCN-Energy | 89.46 | n.a. | 38.86 | n.a. |
| | GKDE-GCN | 89.24 | 80.98 | 39.30 | 30.52 |
| | GPN | 85.72 | 81.56 | 46.12 | 38.98 |
| | Ours | 86.21 | **83.94** | **97.34** | **96.80** |
| Coauthor Physics | APPNP | **93.27** | n.a. | 38.14 | n.a. |
| | VGCN-Energy | 92.86 | n.a. | 37.19 | n.a. |
| | GKDE-GCN | 92.77 | 86.12 | 37.08 | 25.13 |
| | GPN | 91.14 | **89.63** | 41.43 | 35.64 |
| | Ours | 89.93 | 88.83 | **99.14** | **99.10** |
| OGBN Arxiv | APPNP | 77.55 | n.a. | 54.57 | n.a. |
| | VGCN-Energy | **77.89** | n.a. | 54.87 | n.a. |
| | GKDE-GCN | 77.47 | **77.55** | 61.62 | 62.33 |
| | GPN | 75.44 | 72.71 | 55.64 | 52.99 |
| | Ours | 75.30 | 72.85 | **83.95** | **81.54** |

Alea: Aleatoric, Epi.: Epistemic, w/: with propagation

CoraML

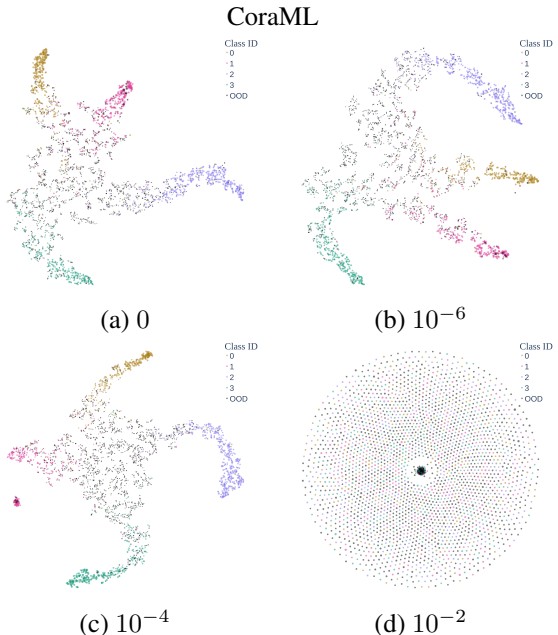

(a) 0           (b) $10^{-6}$

(c) $10^{-4}$           (d) $10^{-2}$

Figure 2: latent representation for CoraML

CiteSeer

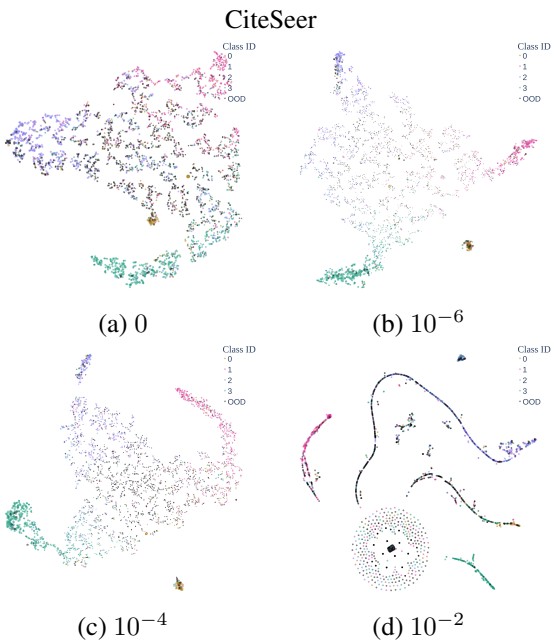

(a) 0           (b) $10^{-6}$

(c) $10^{-4}$           (d) $10^{-2}$

Figure 3: latent representation for CiteSeer

CoauthorCS

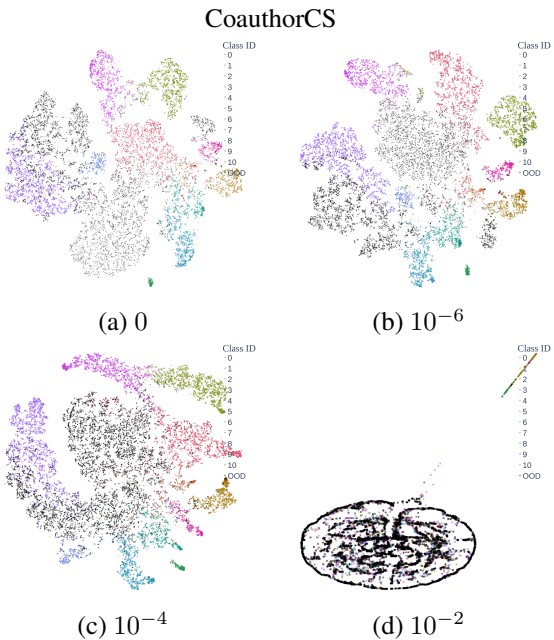

(a) 0                                      (b) $10^{-6}$

(c) $10^{-4}$                              (d) $10^{-2}$

Figure 4: latent representation for Coauthor CS

CoauthorPhysics

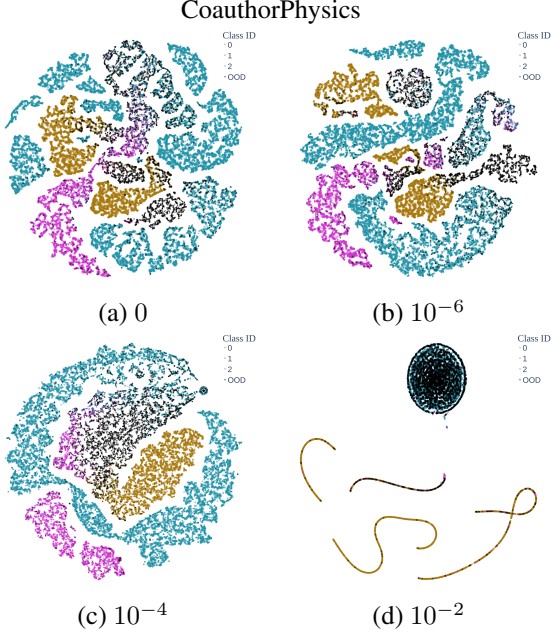

(a) 0                                      (b) $10^{-6}$

(c) $10^{-4}$                              (d) $10^{-2}$

Figure 5: latent representation for Coauthor Physics

## E.4 Graph Activation

In this subsection, we present the t-SNE visualizations of the learned representational space for various datasets in the following figures, without applying distance regularization. Instead, we introduce different activation functions. It is worth noting the notable distinction in quality when using the LogSigmoid activation function, which appears to be the smoothest among the activation functions employed on CiteSeer and Amazon Computers datasets. Once again, the size of each node corresponds to the square root of the learned evidence. Additionally, the color black indicates out-of-distribution (OOD) instances across all datasets, while distinct colors represent different classes.

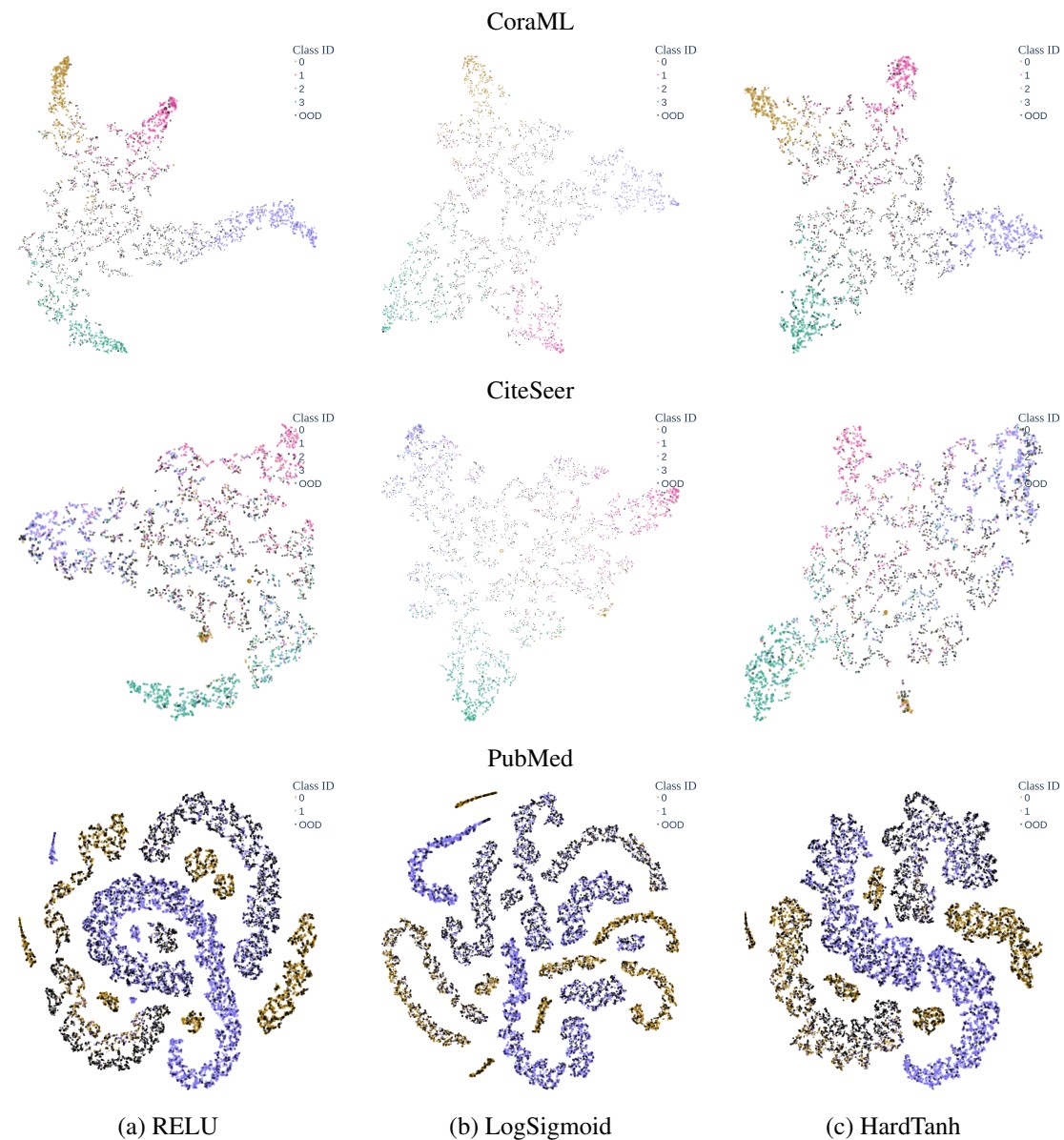

(a) RELU          (b) LogSigmoid          (c) HardTanh

Figure 6: Latent representation for CoraML, CiteSeer and PubMed on different graph activation functions: RELU, LogSigmoid, and HardTanh.

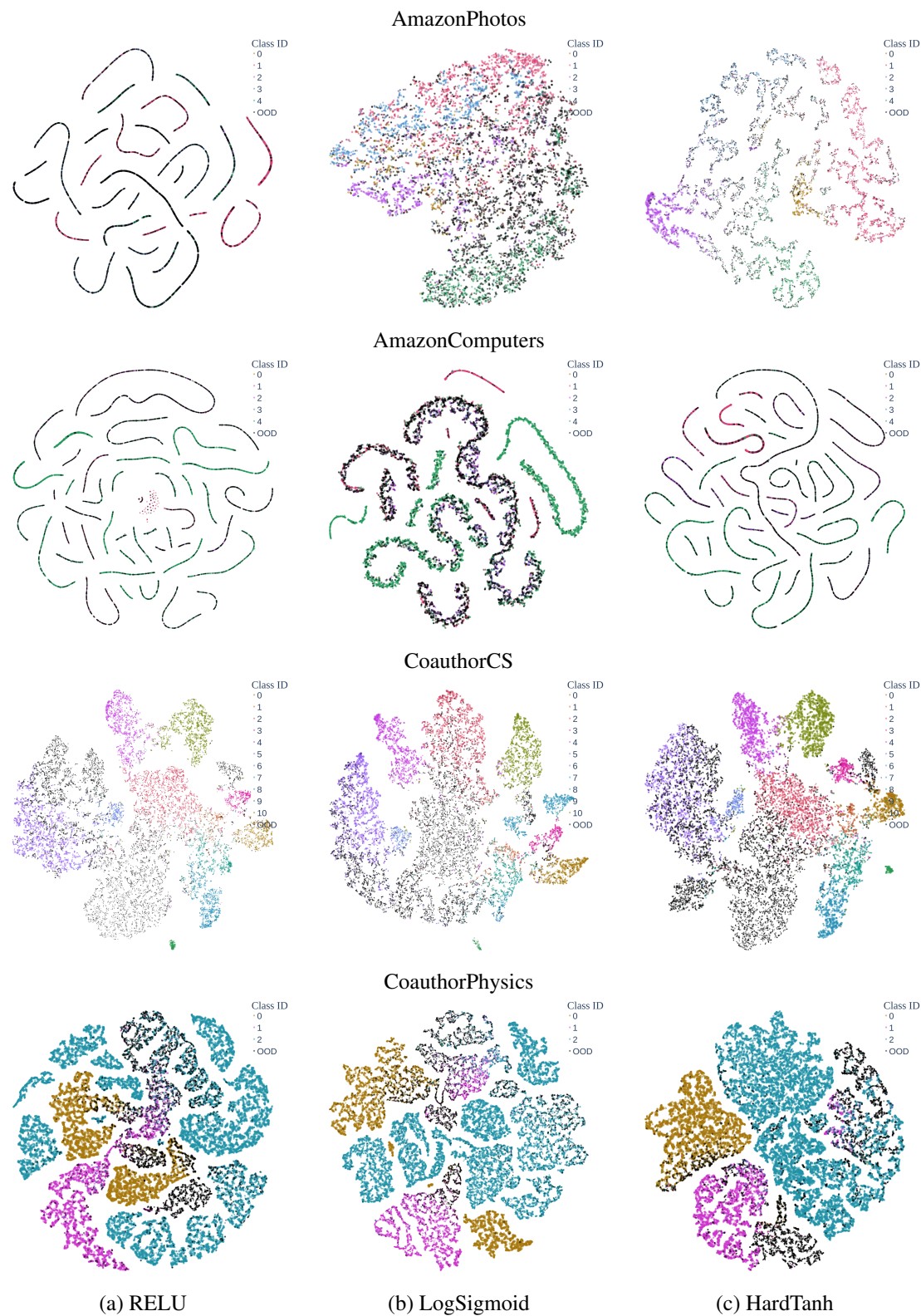

Figure 7: Latent representation for AmazonPhotos, AmazonComputers, CoauthorCS and Coauthor-Physics on different graph activation functions: RELU, LogSigmoid, and HardTanh.

## E.5 Ablation Study

We show the full ablation study on three datasets: CoraML, CiteSeer and PubMed in Table 8.

Table 8: Ablation Study with OOD Detection task (cont.)

| Data | Model | ID-ACC | AUROC | | | AUPR | | |
|------|-------|--------|--------|--------|--------|--------|--------|--------|
| | | | Alea w/ | Epi w/ | Epi w/o | Alea w/ | Epi w/ | Epi w/o |
| CoraML | GPN | 88.51 | 83.25 | 86.28 | **80.95** | 75.79 | 79.97 | 72.81 |
| | GPN-CE | 89.31 | 82.58 | 83.91 | 80.88 | **76.54** | 77.60 | **76.05** |
| | GPN-CE-ACT | 89.87 | 83.34 | 86.96 | 75.60 | 74.96 | 79.74 | 62.73 |
| | GPN-CE-ACT-GD | **90.06** | **83.94** | **87.20** | 76.12 | 76.26 | **80.36** | 63.32 |
| Citeseer | GPN | 69.79 | 72.46 | 70.74 | 66.65 | 55.14 | 50.52 | 44.93 |
| | GPN-CE | 70.98 | 74.20 | 73.75 | 68.41 | 58.12 | 53.55 | 46.60 |
| | GPN-CE-ACT | 71.96 | 74.72 | 77.97 | 72.28 | 60.41 | 56.04 | 50.73 |
| | GPN-CE-ACT-GD | **72.51** | **75.22** | **78.98** | **73.21** | **62.30** | **58.63** | **52.73** |
| PubMed | GPN | **94.08** | 71.84 | 73.91 | 71.2 | 57.92 | 67.19 | 59.72 |
| | GPN-CE | 93.84 | 74.19 | 78.32 | 74.50 | 59.85 | 74.11 | 64.55 |
| | GPN-CE-ACT | 93.84 | 74.19 | 78.32 | 74.50 | 59.85 | 74.11 | 64.55 |
| | GPN-CE-ACT-GD | 93.84 | **75.23** | **81.76** | **77.79** | **60.75** | **78.16** | **69.19** |

[*] Alea: Aleatoric, Epi.: Epistemic, w/: with propagation

GPN is the original results from the GPN paper with default hyperparameters and ReLU as the middle activation function, GPN-CE is the original GPN model with re-tuned dirichlet entropy regularization weight; GPN-CE-ACT is the original GPN model with re-tuned entropy regularization weight and activation function; GPN-CE-ACT-GD/(Ours) add the distance-based regularization term and tuned the two weights and activation function.

