# OpenReview forum: "Improvements on Uncertainty Quantification for Node Classification via Distance Based Regularization"
_NeurIPS.cc/2023/Conference — NeurIPS 2023 poster_

### Official Review · Reviewer_XBDF · 2023-07-05

**Soundness:** 3 good
**Presentation:** 3 good
**Contribution:** 2 fair
**Rating:** 6
**Confidence:** 1

**Summary:**

This paper focuses on uncertainty quantification in node classification. The authors examine the drawbacks of UCE. Inspired by our theoretical analysis, they introduced a distance-based regularization technique to obtain a better representation network for the uncertainty quantification. Experimental results show the superior performance of the proposed approach compared to the current state-of-the-art methods in out-of-distribution (OOD) detection and misclassification detection.


**Strengths:**

1. This paper is well motivated, it discusses the limitations of GPN model on semi-supervised node classification tasks, and it provides detailed theoretical analysis to reveal the limitations of GPN for detecting OOD nodes

2. This paper proposes a distance regularizer to overcome the above limitation.

3. This paper provides extensive experimental results that demonstrate that the proposed method can achieve state-of-the-art performance on both out-of-distribution (OOD) detection and misclassification detection.

**Weaknesses:**

This paper is not in my area, I'm not familiar with existing work about both node classification and uncertainty qualification. So I cannot assess the contribution of this work reliably.

**Questions:**

The authors should better discuss the limitation of their work.

---

> ### Author Rebuttal · Authors · 2023-08-09
>
> Questions: The authors should better discuss the limitation of their work.
>
> We discussed the limitations of Theorem 1 in lines 159 to 165 on Page 4 and the limitations of Theorem 8 in lines 226-234 on Page 6. In addition, our proposed regularization terms proposed in Section 4.2 are most effective for homophily graphs than heterophily graphs. All eight benchmark datasets used are homophily graphs. We will add a section to summarize these limitations when it comes to revision.

---

### Official Review · Reviewer_dzMP · 2023-07-06

**Soundness:** 3 good
**Presentation:** 3 good
**Contribution:** 3 good
**Rating:** 7
**Confidence:** 3

**Summary:**

This manuscript studies the uncertainty quantification for the non-independent node-level prediction problem. The proposed framework is built upon the graph posterior network (GPN) with minimizing uncertainty cross-entropy as the main loss function. A distance-based regularization technique is further proposed to learn representational space mappings in order to handle OOD samples.

**Strengths:**

The topic of quantifying the uncertainty of GNN predictions is timely and interesting. The overall presentation is easy to follow.

I checked the overall theoretical contributions and haven't found major flaws.

The proposed two regularization terms are validated by extensive experiments.

**Weaknesses:**

I am not seeing major drawbacks, except there are some points related to the problem definition.

**Questions:**

1. How can Eq. (1) and (2) derive the aleatoric uncertainty and epistemic uncertainty? The authors do not mention the rationale behind this derivation, which makes it hard to understand the whole process.

**Limitations:**

I am not seeing major drawbacks.

---

> ### Author Rebuttal · Authors · 2023-08-09
>
> Weakness 1 (W1): How can Eq. (1) and (2) derive the aleatoric uncertainty and epistemic uncertainty? The authors do not mention the rationale behind this derivation, which makes it hard to understand the whole process.
>
> The aleatoric and epistemic uncertainty measures defined in Equation 2 are the same measures used in the PN and GPN papers. Please refer to Section 2 of the PN paper and Section 3.3 in the recent TMLR’23 survey paper by Ulmer et al. for more detailed discussions on these Dirichlet-based uncertainty measures. In the following, we summarize some main advantages of these uncertainty measures over the alternative ones.
>
> First, the Dirichlet-based epistemic uncertainty measure can be interpreted as the spread of the Dirichlet distribution about the class probability vectors that are predicted by different model parameters ($\theta$) sampled from the posterior $p(\theta|D)$, where $D$ refers to the training data. GPN and PN directly predict the Dirichlet distribution for each given input $x$ in a deterministic way without explicitly estimating the posterior $p(\theta|D)$. If the spread is large ($u_i^{epis}$ is large), it means the model (epistemic) uncertainty is high (the input sample is distant from the training samples in the feature space).
>
> Second, the aleatoric uncertainty measure ($u_i^{alea}$) is the largest projected class probability (multiplied by -1). If this uncertainty measure is high, it indicates that all the class probabilities are low (the input sample is ambiguous).
>
> Third, the Dirichlet-based aleatoric and epistemic uncertainty measures in Definition 1 can be calculated via a single forward pass. In comparison, other aleatoric measures based on samples of the posterior $p(\theta|D)$, such as dropout-based and deep ensemble-based, require multiple forward passes and are unsuitable for real-time applications.
>
> We will include the aforementioned details of these two uncertainty measures in our revised version to make our paper self-contained.
>
> References:
>
> [1] Ulmer, Dennis, Christian Hardmeier, and Jes Frellsen. "Prior and posterior networks: A survey on evidential deep learning methods for uncertainty estimation." Transactions on Machine Learning Research (2023).
>
> [2] Charpentier, Bertrand, Daniel Zügner, and Stephan Günnemann. "Posterior network: Uncertainty estimation without ood samples via density-based pseudo-counts." Advances in Neural Information Processing Systems 33 (2020): 1356-1367.
>
> [3] Stadler, Maximilian, et al. "Graph posterior network: Bayesian predictive uncertainty for node classification." Advances in Neural Information Processing Systems 34 (2021): 18033-18048.

---

### Official Review · Reviewer_nRfd · 2023-07-07

**Soundness:** 2 fair
**Presentation:** 1 poor
**Contribution:** 3 good
**Rating:** 4
**Confidence:** 2

**Summary:**

The paper proposes a distance-based regularization to improve the OOD detection and misclassification detection tasks on the graph node classification. The regularization enforces the clustered OOD nodes to be close in the latent space.

**Strengths:**

- The paper includes comprehensive experiments on multiple commonly used benchmark dataset and detailed ablation study.

- The paper provides theoretic analysis on the limitation of the GPN.

**Weaknesses:**

- Equation 2 defines aleatoric and epistemic uncertainty. Why are they defined in this way? Is there any reference or reason?

- The writing is not clear. In section 4.1, there are a lot of theorems, but a sentence is missing to summarize what is the limitation of the UCE. Although at the beginning of section 4, there is a summary of those theorems, but it fails to present the overall idea of what the limitation is. Also, it is hard to connect Section 4.1 to 4.2. Why would the limitation of UCE loss mentioned in section 4.1 motivate the distance minimization on the graph?

- The proposed model improves AUPRC but has worse AUROC. It is not easy to tell whether the model is better. How does the model manage to improve AUPRC by 50% while remaining AUROC unchanged in Table 2? How could a model with 95% AUPRC has only 80% AUROC? Could the authors plot out the ROC and PR curves?

**Questions:**

The authors should improve writing in the revision by providing more contexts on the claims and definitions. It is also questionable on the results in the Table 2. I hope the author can clarify that.

**Limitations:**

Yes, the authors addresses limitations in the theoretic analysis.

---

> ### Author Rebuttal · Authors · 2023-08-10
>
> Weakness 1 (W1).
>
> The aleatoric and epistemic uncertainty measures defined in Equation 2 are the same measures used in the PN and GPN papers. Please refer to Section 2 of the PN paper and Section 3.3 in the recent TMLR’23 survey paper by Ulmer et al. for more detailed discussions on these Dirichlet-based uncertainty measures. In the following, we summarize some main advantages of these uncertainty measures over the alternative ones.
>
> First, the Dirichlet-based epistemic uncertainty measure can be interpreted as the spread of the Dirichlet distribution about the class probability vectors that are predicted by different model parameters ( $\theta$ ) sampled from the posterior $p(\theta|D)$, where $D$ refers to the training data. GPN and PN directly predict the Dirichlet distribution for each given input x in a deterministic way without explicitly estimating the posterior $p(\theta|D)$. If the spread is large ($u_i^{epis}$ is large), it means the model (epistemic) uncertainty is high (the input sample is distant from the training samples in the feature space).
>
> Second, the aleatoric uncertainty measure ( $u_i^{alea}$ ) is the largest projected class probability (multiplied by -1). If this uncertainty measure is high, it indicates that all the class probabilities are low (the input sample is ambiguous).
>
> Third, the Dirichlet-based aleatoric and epistemic uncertainty measures in Definition 1 can be calculated via a single forward pass. In comparison, other aleatoric measures based on samples of the posterior p(|D), such as dropout-based and deep ensemble-based, require multiple forward passes and are unsuitable for real-time applications.
>
> We will include the aforementioned details of these two uncertainty measures in our revised version to make our paper self-contained.
>
> References:
>
> Ulmer, Dennis, Christian Hardmeier, and Jes Frellsen. "Prior and posterior networks: A survey on evidential deep learning methods for uncertainty estimation." Transactions on Machine Learning Research (2023).
>
> W2.
>
> The overall idea about the limitations of the UCE loss is that this loss function alone is insufficient to learn a representation space that separates OOD from ID nodes using the GPN model. We discussed the motivation for the distance minimization on the graph based on the limitations of the UCE loss in the first two paragraphs of Section 4.2 on page 6. We used Example 2 on Page 4 to illustrate that when OOD nodes are separable from the ID nodes in the feature space based on OOD-specific features, they can be close in the learned representation space, as OOD-specific features are discarded. In other words, the learned representation space by GPN based on the UCE loss is not guaranteed to preserve the distance between OOD and ID nodes in its representation learning step. We consider distance regularization proposed in Section 4.2, as it helps prevent the model from discarding relevant OOD-specific events while decreasing variation between nodes in the representation space.
>
> We will improve the writing of Section 4 with implications of each theorem and connections among these theorems in our revised version.
>
> W3.
>
> (1) Table 2 reports the empirical results on misclassification detection. We want to highlight that our main contributions are on OOD detection but not misclassification detection. As stated at the end of Section 1 (Introduction), our first contribution is theoretical evaluations of the limitations of the UCE loss for OOD detection. Our second contribution is our distance-based regularization which helps prevent the model from discarding relevant OOD-specific features. In the third contribution, our empirical comparisons on eight datasets, as shown in Table 1 (three datasets) in the main paper and Table 3 (five datasets) in the complementary material, have well demonstrated that our method outperformed the GPN baseline and four other baselines in the majority of the settings on both the AUROC and AUPR metrics.
>
> Similarly, the main contributions of the GPN paper are also OOD detection but not misclassification detection. For example, the results in Table 10 and Table 11 in the complementary material of the GPN paper demonstrated that for the misclassification detection task,  GPN performed worse than the best of the baselines for seven of the eight datasets in the range between 0.5\% and 3.7\% of the AUROC metric. GPN performed worse than the best of the baselines for four of the eight datasets in the range between 0.1\% and 21.7\% based on the AUPR metric.
>
> (2) For misclassification detection, our empirical results demonstrate that our method outperformed the baselines for all the datasets based on the AUPR with large percentages. Based on the AUROC metric, our method performed worse than the best of the baselines, but the differences are within around 3\% for six of the eight datasets: Amazon Computers, Amazon Photos, Coauthor CS, Coauthor Physics, and ODBG Arxiv, and PubMed.
>
> See our response to W2 of Reviewer Spkg for more details for two related clarifications: (a) the GPN (the main competitor in our paper) has the same empirical behavior compared to their baselines as reported in the GPN paper. (b) Some papers report similar empirical behavior in classification tasks, and we gave interpretations for this empirical behavior.
>
> (3) About our model learned based on the PubMed dataset that has 95\% AUPR but only 80\% AUROC, we have plotted out the ROC and PR curves of our method for this dataset and two other datasets: CoraML and CiteSeer, which can be referred to the pdf in the global rebuttal.
>
> Questions.
>
> Thanks for your suggestions. We will improve our presentation with more context on assumptions, definitions, and implications.
>
> We have explained the results in Table 2 with more details to clarify the potential confusion (e.g., the observation that our method gives higher AUPR but lower AUROC for misclassification detection). We will add these details in the revision.

---

> > ### Comment · Reviewer_nRfd · 2023-08-15
> >
> > I would like to thank the authors response. I am not in this area, so it is quite hard to follow the original paper. The response is helpful to help me understand. I increased my score and suggest the authors improving writing.

---

> > > ### Author Response · Authors · 2023-08-18
> > >
> > > Thanks for increasing your score and for your suggestions. We will add more context to the claims and definitions in the main paper. We will also extend the complementary material to introduce the related concepts in more detail to make our paper more self-contained.

---

### Official Review · Reviewer_Spkg · 2023-07-09

**Soundness:** 3 good
**Presentation:** 3 good
**Contribution:** 3 good
**Rating:** 8
**Confidence:** 3

**Summary:**

## Post Rebuttal Update

I have engaged with the authors for the rebuttal, and found their responses informative, prompting me to increase my score from a 7 to an 8.

## Original Review
The paper proposes theoretical results for Graph Posterior Networks (GPNs) that use the uncertainty cross-entropy loss (UCE), and shows that it is possible to trivially set the uCE loss to zero under certain conditions. The paper then proposes a way to regularise the UCE loss using distance, ensuring that OOD nodes that are clustered together are clustered together in latent space.

**Strengths:**

I like the theoretical analysis section of this paper. It constructs a way of obtaining zero UCE for a pathological example, showing that there are improvements required to be made to the GPN+UCE framework. Furthermore, it incorporates the fact that OOD points are not explicitly modeled in the UCE framework, and suggests ways to incorporate this. I also find the baselines considered pretty comprehensive, and the authors have done a good job covering a range of techniques. I also like the further ablations the paper has, such as selecting activation functions as well, using a validation CE loss, and I like that this has been properly ablated against. In general, I found the paper well motivated and written. I went through the proofs of Theorem 1-6 and found them accurate to my admittedly limited knowledge.

**Weaknesses:**

I have a few criticisms of the paper that I would like to get addressed -

1. I would like a thorough description of all the assumptions made for Theorem 1, and where these assumptions would be violated or not, maybe with a toy example. For example, I'm not sure how strict the assumption "If the underlying distribution of feature vectors belonging to class k, denoted by $X_k$ disjoint to each other," is, and would like a discussion of the drawbacks/limitations of the assumptions made in Theorem 1.
2. Is the ball $B\left(\mathbf{z}_k, r_k\right)$ guaranteed to have finite volume necessarily? There are pathological behaviours in neural networks that use ReLU activations, where it is possible to obtain high softmax scores (i.e. class probabilities) arbitrarily far from the training manifold [1], and this would make the volume of the ball very large potentially in some cases.


[1] https://arxiv.org/abs/1812.05720

**Questions:**

In Table 2, it seems like the regularisation consistently gives lower AUROC but higher AUPR compared to the GPN baseline. Can you explain why this might be the case?

**Limitations:**

N/A.

---

> ### Author Rebuttal · Authors · 2023-08-09
>
> Weakness 1 (W1).
>
> We have two assumptions in Theorem 1: (1) the underlying distribution of feature vectors belonging to class $k$, denoted by $X_k$,  is disjoint to each other; (2) the multi-layer perception (MLP) module, $f_\theta$,  and the normalizing flow module, $h_\phi$, in the posterior network, can be arbitrarily complex. The first assumption means that the supports of the in-distribution (ID) classes are separable in the feature space, where the support of a class is defined as the region in the feature space that has non-zero density for this class. The second assumption means that the posterior network defined based on $f_\theta$ and $h_\phi$ is a universal approximator. The universal approximation theorem for MLP with a single hidden layer has been studied in the literature (e.g., Pinkus’1919) and used in the theoretical analysis of graph neural networks and other deep learning models (e.g., Loukas’2020).
>
> An implication of Theorem 1 is that the UCE loss is insufficient to separate OOD from ID nodes. We present Example 2 to discuss this implication (lines 152 to 158). We discussed the limitations based on our second assumption in lines 159 to 165 and gave an example to show that our analysis is still informative about the shape these networks are likely to take. Our first assumption may not hold in datasets where some ID classes are not separable due to noises on features and/or class labels. However, our intuition is that if the UCE loss is insufficient to separate OOD from OOD nodes in the separable case, this loss function will more likely fail in the more challenging non-separable case. We will consider the theoretical analysis based on the relaxation of the two assumptions in our future work. For example, we may get similar results in Theorem 1 by allowing a certain level of noise on class labels.
>
> W2 (part 1)
>
> Yes, here we assume that f is arbitrarily complex (no requirements on it being a ReLU-based MLP) so we choose the ball to have a finite volume. The theoretical evaluations will be more challenging when the support regions are not bounded. For this reason, some papers related to OOD analysis (e.g., Karthik et al. 2021) directly assume that the support regions are bounded.
>
> W2 (part 2)
>
> We mainly look at the misclassification results using aleatoric uncertainty, which is the most effective uncertainty measure for misclassification detection. There are three datasets examined in Table 2, and 5 additional datasets in Table 4 in the complementary material. It is true that our method gives lower AUROC, but we think our results are comparable to GPN for five out of the eight datasets in the sense that the differences are between 0.14\% and 1.5\%. For the remaining three datasets, i.e., CoraML, Citeseer, and Amazon Photos, the differences are a bit larger, between 2.8\% and 6.7\%. On the other hand, our method achieves much higher AUPR scores than GPN for seven of the eight datasets: the improvements are between 28.31\% and 54.67\% with an average of 43.18\% improvement. For the  Citeseer dataset, our method is better than GPN for 11.89\%.
>
> First, we would like to point out that the GPN paper reported a similar behavior for comparison to their baseline models in that it gives lower AUROC but higher AUPR; please refer to the misclassification detection results in Table 10 and Table 11 of the GPN paper. For example, Table 10 of the GPN paper shows that the GKDE-GCN method has the highest AUPR (49.61\%) but has a lower AUROC (80.80\%) than six of the seven baselines for the CiteSeer dataset. For the PubMed dataset, GPN has the highest AUPR (40.74\%) but has a lower AUROC (80.46\%) than six of the seven baselines.
>
> Second, we explain why it makes sense to have higher AUPR but lower AUROC. These two metrics offer different perspectives to measure the quality of a ranking on data points for separating positives and negatives. A lower AUROC but a higher AUPR for our method, compared to GPN, indicates that our method produces more true positives among the top-ranked nodes than GPN, while GPN can separate true positives and negatives better than our method among the lower-ranked nodes. We note that a high AUPR is important for applications where human experts want to manually verify the top-ranked misclassified nodes or anomalies. Due to the high cost of manual verification, it is important to ensure a high rate of true positives among the top-ranked data points, and in this case, AUPR may be a better metric than AUROC.
>
> Lastly, not just for misclassification detection, this empirical behavior also exists for classification tasks as well. For example, Jesse and Goadric pointed out in their 2006 ICML paper that algorithms that optimize AUROC are not guaranteed to optimize AUPR and vice versa. In addition, a recent arXiv’23 paper by Yuan et al. reports similar empirical observations for classification tasks. For example, some learned classifiers had the highest AUPR (4\% to 10\% higher than the other methods) but the lowest AUROC (8\% to 10\% lower than the other methods) in the SOAP dataset (See Table 2 and the second paragraph from the last in Section 4.1 of the arXiv’23 paper).
>
> References:
>
> [1] Pinkus, Allan. "Approximation theory of the MLP model in neural networks." Acta numerica 8 (1999): 143-195.
>
> [2] Loukas, Andreas. "How hard is to distinguish graphs with graph neural networks?." NeurIPS (2020): 3465-3476.
>
> [3] Ahuja, Kartik, et al. "Invariance principle meets information bottleneck for out-of-distribution generalization." NeurIPS (2021): 3438-3450.
>
> [4] Davis, Jesse, and Mark Goadrich. "The relationship between Precision-Recall and ROC curves." ICML (2006): 233-240.
>
> [5] Yuan, Zhuoning, Dixian Zhu, Zi-Hao Qiu, Gang Li, Xuanhui Wang, and Tianbao Yang. "Libauc: A deep learning library for x-risk optimization." arXiv preprint arXiv:2306.03065 (2023).

---

> > ### Comment · Reviewer_Spkg · 2023-08-19
> >
> > Thanks for engaging with my questions. It would be nice to add a small paragraph summarizing the assumptions you mentioned as a response to W1. I also appreciate the AUPR curves, and like the discussion about where it is important to prioritize a model with high AUPR and lower AUROC and vice versa. I've increased my score to an 8.

---

> > > ### Author Response · Authors · 2023-08-19
> > >
> > > Thanks for increasing your score and for your suggestions. Yes, we will add a small paragraph summarizing the assumptions we mentioned as a response to W1 in the revision. This will help future readers better understand our theoretical evaluations.

---

### Author Rebuttal · Authors · 2023-08-10

We append the ROC-PR curves for our model related to Table 2 in the attached pdf.

---

### Decision · Program_Chairs · 2023-09-21

**Decision:**

Accept (poster)

**Comment:**

The paper presents theoretical results on Graph Posterior Networks (GPNs) with uncertainty cross-entropy loss (UCE) and highlights its limitations. To address these limitations, a distance-based regularization is introduced to ensure that clustered OOD nodes remain close in the latent space. This is argued to improve OOD detection and graph node classification.

### Strengths:

- Strong theoretical foundation with an analysis that reveals the limitations of the GPN+UCE framework.
- Proposes a method to address the fact that OOD points are not explicitly modeled in the UCE framework.
- Comprehensive experimentation on multiple benchmark datasets with detailed ablation studies.
- Demonstrates state-of-the-art performance on OOD detection and misclassification detection tasks.

### Weaknesses:

- Multiple reviewers found the writing unclear, particularly regarding 1. the linkage between sections and the overall presentation of theorems; and 2. the derivation of aleatoric and epistemic uncertainties (Eqs. 1-2).
- Concerns were raised about certain assumptions made in Theorem 1, specifically their limitations or instances where they might be violated.
- Questions were raised regarding the results in Table 2, especially the inconsistency between AUROC and AUPR scores. The model manages to improve AUPRC significantly while keeping AUROC largely unchanged.

Weighting the strengths and weaknesses, the majority of reviewers find value in the presented work. Given the significance of the work, I recommend accept. It is essential for the authors to address the raised concerns during the revision process.